# SELECTIVE LoRA FOR DOMAIN-ALIGNED DATASET GENERATION IN URBAN-SCENE SEGMENTATION

## ABSTRACT

This paper addresses the challenge of data scarcity in semantic segmentation by generating datasets through fine-tuned text-to-image generation models, reducing the costs of image acquisition and labeling. Segmentation dataset generation faces two key challenges: 1) aligning generated samples with the target domain and 2) producing informative samples beyond the training data. Existing methods often overfit and memorize training data, limiting their ability to generate diverse and well-aligned samples. To overcome these issues, we propose Selective LoRA, a novel fine-tuning approach that selectively identifies and updates only the weights associated with necessary concepts (*e.g.* style or viewpoint) for domain alignment while leveraging the pretrained knowledge of the image generation model to produce more informative samples. Our approach ensures effective domain alignment and enhances sample diversity. We demonstrate its effectiveness in generating datasets for urban-scene segmentation, outperforming baseline and state-of-the-art methods in in-domain (few-shot and fully-supervised) settings, as well as domain generalization tasks, especially under challenging conditions such as adverse weather and varying illumination, further highlighting its superiority.

## 1 INTRODUCTION

The amount of labeled data is crucial for achieving high performance in semantic segmentation. However, acquiring diverse image samples, especially in rare or complex scenarios, and providing pixel-wise annotations are labor-intensive and time-consuming. Recent advances in text-to-image generation models (T2I models) (Rombach et al., 2022; Saharia et al., 2022; Podell et al., 2023) have significantly improved the image quality, enabling their use in data creation for perception tasks such as segmentation with minimal human effort. Existing studies (Zhang et al., 2021; Baranchuk et al., 2022; Wu et al., 2023a;b) leverage these models, such as Stable Diffusion (Rombach et al., 2022), pretrained on large-scale datasets (Schuhmann et al., 2022). Utilizing rich generative features from the T2I model, label generation modules can be trained with minimal labeled data, effectively parsing semantic regions. Furthermore, these models also provide controllability through text input, allowing for the generation of underrepresented distributions. These approaches have proven particularly effective in augmenting labeled datasets (Zhang et al., 2021; He et al., 2022a; Azizi et al., 2023; Wu et al., 2023a), addressing data scarcity, and creating desired distributions, such as long-tailed class distributions (Shin et al., 2023) or adverse weather condition (Jia et al., 2023).

There are two primary challenges in generating segmentation datasets using image generation models: 1) aligning with the target domain, and 2) generating informative datasets beyond existing training data. However, previous approaches have overlooked key aspects of these issues. Existing methods (Zhang et al., 2021; Li et al., 2022; Baranchuk et al., 2022; Park et al., 2023) typically produce domain-aligned but non-informative samples, as they train image generation models from scratch using only segmentation datasets, resulting in images that closely resemble the training data. To overcome this, leveraging T2I models such as Stable Diffusion, pretrained on large-scale datasets, is crucial for generating more diverse and informative samples. However, recent methods (Wu et al., 2023a;b; Yang et al., 2024; Jia et al., 2023) often use these pretrained models without fine-tuning for segmentation tasks, leading to poor alignment with the target domain.

Full fine-tuning or Low-Rank Adaptation (LoRA) (Hu et al., 2022) of pretrained T2I models are potential solutions to the two challenges discussed above. However, as shown in Fig. 1, even LoRA fine-tuning often overfits and memorizes the training data, limiting the generation of informative

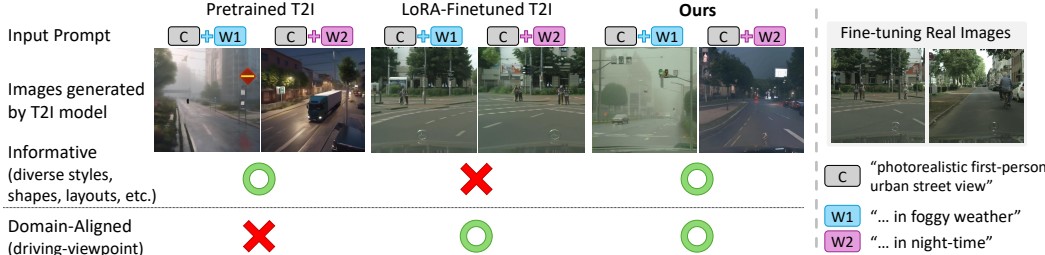

Figure 1: Motivation of our paper: Pretrained T2I models can generate informative images but often struggle with viewpoint alignment. LoRA fine-tuning on Cityscapes enables T2I models to generate driving-viewpoint images but leads to overfitting to the Cityscapes style and content. We aim to exclusively learn the desired concept (*e.g.*, viewpoint) from the source dataset for generating domain-aligned and informative samples.

samples beyond the training dataset because the model learns every concept (*e.g.*, viewpoint, style, object shape, layout, etc.) present in the training data. Therefore, we need a method to selectively learn only the necessary concepts (*e.g.*, viewpoint or style) for aligning with the target domain, while leveraging the pretrained knowledge of the T2I model to generate more informative samples.

The target domain can vary depending on the problem settings (*e.g.*, in-domain or domain generalization), which means the necessary concepts for fine-tuning also vary. In in-domain settings, the model needs to learn the style from the training data. However, in domain generalization (DG) settings, where the model is evaluated on an unknown target domain, it is more beneficial to learn only the viewpoint. For example, when the target domain is ACDC (Sakaridis et al., 2021), which includes driving-scene viewpoints and adverse weather, but the training dataset is Cityscapes (Cordts et al., 2016), which consists of driving-scene viewpoints with only clear-day conditions, a diverse weather conditional dataset (informative) combined with a driving-viewpoint (domain-aligned) could serve as the optimal dataset for the problem. However, as shown in Fig. 1, pretrained model often fails to generate a driving viewpoint while the LoRA fine-tuned model generates only the clear-day style of the Cityscapes, even if 'foggy' or 'night-time' conditions are added as text prompts, highlighting the need for a method that can selectively learn only the viewpoint from Cityscapes training data.

To the best of our knowledge, our research is the first to comprehensively address these issues. We propose Selective LoRA that identifies and updates only the weights related to desired concepts (*e.g.*, viewpoint or style) while preserving the rest to leverage the knowledge of pretrained T2I models. This approach enables the model to effectively capture and learn the specific concepts necessary for aligning with the target domain, resulting in generated images that are not only well-aligned but also more diverse and informative. For instance, if the desired concept is driving-scene viewpoints, the model learns that viewpoint alone and generates images that extend beyond the original training data by incorporating various styles, object shapes, layouts, etc. Additionally, the model's text controllability allows for generating specific styles from user input, making it highly effective in DG settings (Choi et al., 2021; Peng et al., 2022; Lee et al., 2022a; Zhong et al., 2022; Hoyer et al., 2022b;a), such as those requiring diverse conditions like adverse weather or varying illumination.

We demonstrate the effectiveness of our approach in urban-scene segmentation, comparing it to baselines (Hoyer et al., 2022a;b) and other dataset generation methods (Wu et al., 2023a; Jia et al., 2023) in both few-shot and fully-supervised settings, as well as in DG setting.

Our contributions are threefold:

1. We propose Selective LoRA, a novel fine-tuning method that selectively identifies and updates only the weights related to the necessary concepts (*e.g.*, viewpoint or style) for domain alignment, reducing overfitting and preserving pretrained knowledge.

2. Applying Selective LoRA to T2I models generates well-aligned and informative datasets beyond existing training data. This addresses data scarcity by generating image-label pairs from underrepresented distributions (*e.g.*, adverse weather), improving segmentation tasks.

3. Our method demonstrates state-of-the-art performance across various tasks, with improvements of +2.30 mIoU in few-shot, +1.34 mIoU in fully supervised settings on Cityscapes, and +1.53 mIoU in DG benchmarks (ACDC, Dark Zurich, BDD100K, Mapillary Vistas). It consistently generates higher-quality image-label pairs compared to existing methods.

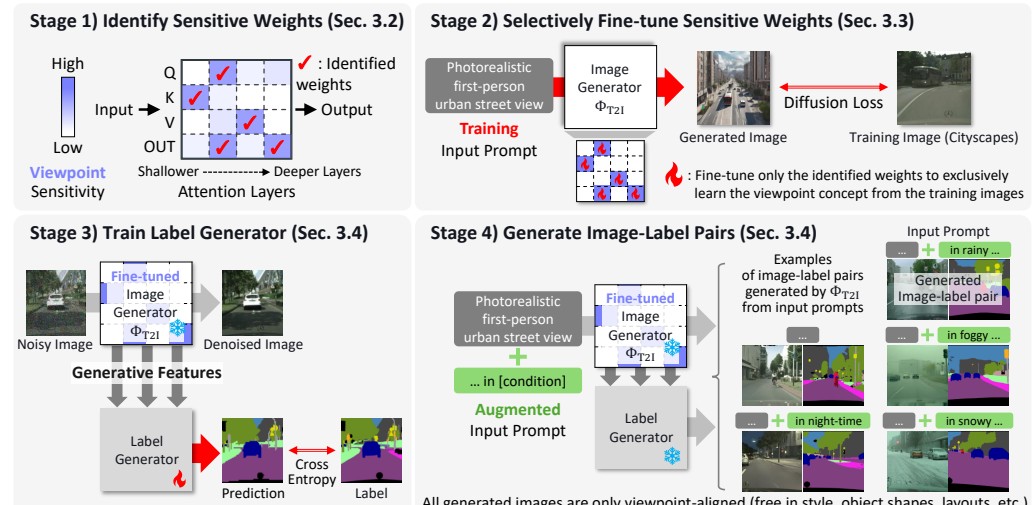

Figure 2: Overview of the proposed framework for generating an urban-scene segmentation dataset by learning the Cityscapes viewpoint. (Stage 1, Section 3.2) we identify sensitive layers to the specific concept (*e.g.*, style, viewpoint). (Stage 2, Section 3.3) we selectively fine-tune the identified sensitive layers using LoRA to learn only the specific concept. (Stage 3, Section 3.4) to produce a corresponding segmentation map, we train a label generator that takes generative features from the T2I model. (Stage 4, Section 3.4 we generate diverse image-label pairs with textual augmentation based on the problem settings (*e.g.*, in-domain, domain generalization).

## 2 RELATED WORK

### 2.1 TEXT-TO-IMAGE GENERATION AND PARAMETER-EFFICIENT FINE-TUNING

Recent advancements in diffusion architectures (Ho et al., 2020; Rombach et al., 2022) and large-scale image-text dataset (Schuhmann et al., 2022) have enabled high-quality text-to-image generation models (T2I models) (Saharia et al., 2022; Ramesh et al., 2022; Podell et al., 2023; Esser et al., 2024). The quality of images generated by these models has led researchers to personalize them to produce specific concepts or styles(Ruiz et al., 2023; Gal et al., 2022). To achieve better customization, parameter-efficient fine-tuning (PEFT) methods (Hu et al., 2022; Liu et al., 2024; Hayou et al., 2024; Kopiczko et al., 2023; Ding et al., 2023; He et al., 2022b) have been proposed.

While existing PEFT methods aim to prevent overfitting and enable efficient training, they struggle to disentangle irrelevant concepts during fine-tuning, as they may still equally affect all layers. Thus, several studies (Guo et al., 2019; Choi et al., 2022; Lee et al., 2022b) have shown that fine-tuning manually selected layers outperforms full fine-tuning, especially with smaller datasets. Additionally, recent work on Stable Diffusion (Wang et al., 2024; Xing et al., 2024; Basu et al., 2024) identifies control blocks for specific visual attributes by ablating each block manually. In contrast, our approach automates this process, enabling more precise and fine-grained updates to only the most crucial weights, leading to more efficient fine-tuning.

### 2.2 SEGMENTATION DATASET GENERATION

Generating segmentation datasets is challenging due to the need for pixel-wise annotations (Zhang et al., 2021; Li et al., 2022; Baranchuk et al., 2022; Park et al., 2023). To generate segmentation maps for the generated images, segmentation dataset generation frameworks typically use generative features from T2I models as input to the label generator, as shown in Fig. 2 (Stage 3). By leveraging these rich generative features, the label generator requires minimal labeled data, particularly for parsing semantic regions. However, when T2I models are trained from scratch with only the provided segmentation datasets, they often produce non-informative outputs resembling the training data.

More recently, several studies have focused on leveraging the extensive prior knowledge embedded in pretrained T2I diffusion models (Wu et al., 2023a;b; Nguyen et al., 2024; Benigmim et al., 2023; Gong et al., 2023). However, they often overlook the alignment of the generated images with the target domain (*e.g.*, style, viewpoint). In this paper, we investigate the impact of fine-tuning T2I models for segmentation dataset generation, with a focus on ensuring better alignment.                    FIX

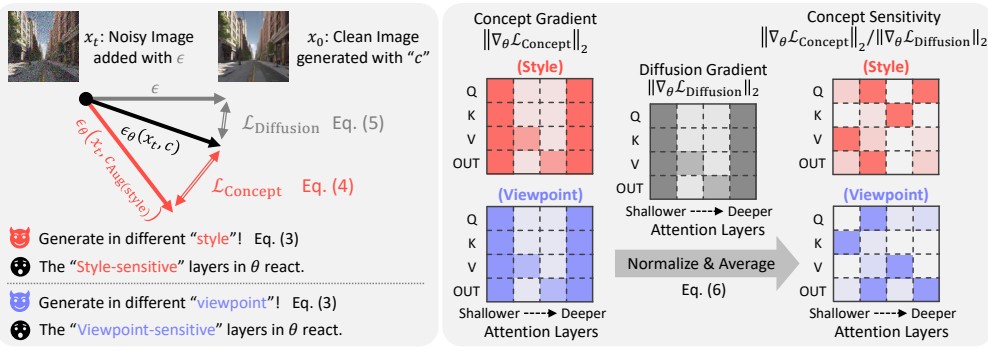

(a) Calculate Concept and Diffusion Loss      (b) Compute Concept Sensitivity

Figure 3: Overview of measuring concept sensitivity. (a) We design the concept loss ($\mathcal{L}_{\text{Concept}}$) with the concept-augmented captions ($c_{\text{Aug}}$), and the original diffusion loss ($\mathcal{L}_{\text{Diffusion}}$) with the added noise $\epsilon$. The concept-augmented captions can be changed according to the desired concept (*e.g.*, style, viewpoint). (b) While each concept gradient represents the reaction of the concept, it has to be normalized with the original diffusion gradient to assess the increased ratio of each layer.

## 3 METHOD

### 3.1 OVERALL FRAMEWORK

Our proposed dataset generation framework for urban-scene segmentation is shown in Fig. 2. The process begins with identifying sensitive weights according to the desired concept in the source dataset (Stage 1), as illustrated in Section 3.2. Next, we selectively fine-tune the top-$k\%$ sensitive weights of the text-to-image (T2I) generation model using LoRA (Stage 2), as discussed in Section 3.3. Then, a label generator is trained with a labeled dataset by using rich generative features from the T2I model as input (Stage 3), as described in Section 3.4. Finally, diverse image-label pair datasets are generated by modifying textual conditions based on the problem settings (in-domain or domain generalization) (Stage 4).

### 3.2 IDENTIFYING SENSITIVE WEIGHTS TO THE DESIRED CONCEPT

To identify the sensitive weights for a specific concept (*e.g.*, style, viewpoint), we first design an objective function called Concept loss ($\mathcal{L}_{\text{Concept}}$) which can be flexibly changed according to the desired concept. As shown in Fig. 3 (a), Concept loss is provided to the noisy image to enforce modifying the concept, such as style or viewpoint of the generation process. For example, when the Concept loss forces the T2I model to modify the style (*e.g.*, photorealistic $\rightarrow$ sketch), the gradient of the Concept loss can be used to identify style-sensitive weights.

For the Concept loss input, we prepare a few generated images $x_0$ with the pretrained T2I model $\Phi_{\text{T2I}}$ parameterized by $\theta$ and the original text prompt $c$. Random Gaussian noise $\epsilon$ is added to the generated images based on the pre-defined timestep $t$ and the timestep scheduling coefficient $\bar{\alpha}_t$. [1]

$$x_0 = \Phi_{\text{T2I}}(c; \theta), \qquad x_t = \sqrt{\bar{\alpha}_t}x_0 + \sqrt{1 - \bar{\alpha}_t}\epsilon, \qquad \epsilon \sim \mathcal{N}(0, \mathbf{I}) \tag{1}$$

Then, we prepare the simply augmented prompts of the original text prompt ($c$) according to style ($c_{\text{Aug(Style)}}$) and viewpoint ($c_{\text{Aug(Viewpoint)}}$).

$$c = \text{"Photorealistic first-person urban street view"} \tag{2}$$

$$
c_{\text{Aug(Style)}} \in
\left\{
\begin{array}{l}
\text{"Sketch of first-person urban street view",} \\
\text{"Watercolor of first-person urban street view",} \\
\text{"Pop-art of first-person urban street view"}
\end{array}
\right\}
\quad
c_{\text{Aug(Viewpoint)}} \in
\left\{
\begin{array}{l}
\text{"Photorealistic urban street in top-down view",} \\
\text{"Photorealistic urban street in high angle view",} \\
\text{"Photorealistic urban street in low angle view"}
\end{array}
\right\}
\tag{3}
$$

We then use the denoising prediction with the augmented captions as pseudo-ground truth, guiding the modification of the image based on a specific concept. Concept loss ($\mathcal{L}_{\text{Concept}}$) is defined by the following equation, similar to the original diffusion loss.

$$\mathcal{L}_{\text{Concept}} := \left\| \epsilon_\theta(x_t, c) - \text{sg}[\epsilon_\theta(x_t, c_{\text{Aug}})] \right\|_2, \tag{4}$$

---

[1] While we add a random noise with a DDPM (Ho et al., 2020) scheduler, any scheduler can be used for this process. Additionally, the $x_0$ will be replaced to latent for the latent diffusion models (Rombach et al., 2022).

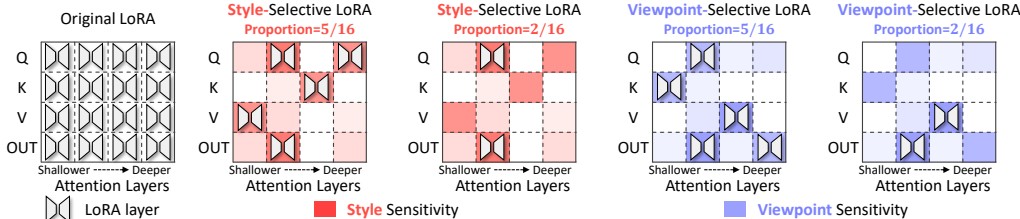

Figure 4: Illustration of Selective LoRA. Unlike the original LoRA method, which applies LoRA layers to all projection layers, we selectively attach LoRA layers to high-sensitivity projection layers based on the desired concept.

where sg indicates the stop-gradient operation. Initially, we calculate the gradient of concept loss ($\|\nabla_\theta \mathcal{L}_{\text{Concept}}\|_2$) to capture the sensitivity of the concept. However, we observe critical bias in the gradients of each layer depending on its position, as discussed in Appendix A.4. To address this, we scale the gradient magnitude for each layer. We calculate the ratio of the gradients between the concept loss and the original diffusion loss ($\|\nabla_\theta \mathcal{L}_{\text{Diffusion}}\|_2$), referred to as *concept sensitivity*. FIX

$$\mathcal{L}_{\text{Diffusion}} := \|\epsilon_\theta(x_t, c) - \epsilon\|_2 \tag{5}$$

$$\text{Concept Sensitivity}(\theta) := \mathbb{E}_{x_0, \epsilon, c_{\text{Aug}}} \left[ \frac{\|\nabla_\theta \mathcal{L}_{\text{Concept}}\|_2}{\|\nabla_\theta \mathcal{L}_{\text{Diffusion}}\|_2} \right] \tag{6}$$

As illustrated in Eq. 6, we average the ratio between the gradients across the generated images ($x_0$), added noise ($\epsilon$), and the augmented prompts ($c_{\text{Aug}}$). While the concept sensitivity can be grouped in various ways, we conduct it in multi-head projection-wise, which we illustrate in Appendix A.3. In summary, the proposed concept sensitivity identifies sensitive layers to the desired concept, which can rapidly learn the target concept by leveraging the reaction of changing the ground truth.

### 3.3 SELECTIVE LoRA

Based on the identified sensitive weights to desired concepts in Section 3.2, we propose a novel parameter-efficient fine-tuning method, which can selectively update only the high-sensitivity layers to maintain the prior knowledge of pretrained T2I models of the other concepts. To achieve this, we start with the Low-Rank Adaptation (LoRA) (Hu et al., 2022), which effectively adapts all the attention layers in the pretrained T2I model with the source dataset. However, while LoRA FIX parameter-efficiently fine-tunes large-scale models, it cannot specify target learning concepts (*e.g.*, style or viewpoint) from the source datasets. Additionally, the follow-up studies of LoRA also have NEW focused on improving the LoRA adapter itself (*e.g.*, architectures) (Wu et al., 2024; Zhao et al., 2024; Renduchintala et al., 2024), little has been explored for identifying which layers are effective for LoRA fine-tuning to learn a specific target concept, especially in urban-scene segmentation.

In contrast, we propose *Selective LoRA*, which selectively adapts a subset of the pretrained layers. As shown in Fig. 4, we select top $k\%$ weights of the entire pretrained model, which will be LoRA fine-tuned based on the concept sensitivity scores (Eq. 6). The key distinction of Selective LoRA lies FIX in *selectively fine-tuning only the crucial layers* based on an automatically computed score, termed *concept sensitivity*, for the desired concept in the source dataset, while previous LoRA studies update all projection layers. The adapted layers and selected ratios can be adjusted based on the concept, allowing for increased control, as illustrated in Fig. 4. In the following sections, we refer to Style-Selective LoRA and Viewpoint-Selective LoRA as Selective LoRA fine-tuning methods based on style and viewpoint sensitivity, respectively.

### 3.4 TRAINING LABEL GENERATOR AND GENERATING DIVERSE SEGMENTATION DATASETS

**Training Label Generator** We train an additional lightweight label generator to produce a seg- NEW mentation label corresponding to the image, following DatasetDM (Wu et al., 2023a). To train the label generator, we add noise to the given labeled image and denoise the image with the fine-tuned T2I model, which can provide semantically rich intermediate multi-level feature maps and cross-attention maps. Then, the label generator receives the feature maps and cross-attention maps as input to predict the label map, as illustrated in Stage 3 of Fig. 2. Distinct from DatasetDM, we train the label generator based on the fine-tuned T2I model using Selective LoRA. The added fine-tuning process causes a significant difference in image-label alignment, which we discussed in Appendix A.6. Furthermore, due to the difference between the base T2I model, architecture details slightly changed as described in Appendix A.1.

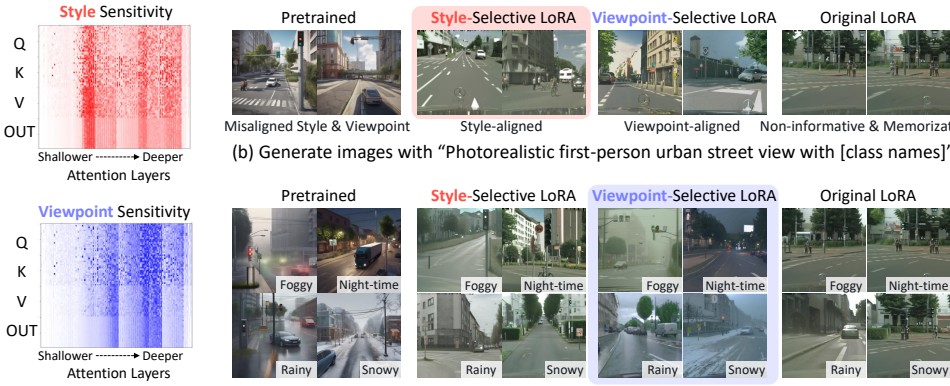

Figure 5: (a) Visualization of the concept sensitivity across the attention layers. We highlight style- and viewpoint-sensitive weights in red and blue, respectively. Each column represents the layer position, ranging from shallow to deep, while each row corresponds to the multi-head projection layers for query, key, value, and output in the attention module. (b, c) Qualitative comparison between the pretrained T2I model and fine-tuned models using original LoRA, Style-Selective LoRA, and Viewpoint-Selective LoRA. The pretrained model often misaligns with the viewpoint and style of the source domain, while the original LoRA memorizes the training examples. In contrast, the proposed Style- and Viewpoint-Selective LoRA selectively learn style and viewpoint concepts from the source dataset (Cityscapes), respectively.

**Generating Diverse Dataset**   Lastly, we introduce the diverse image-label pair generation process by modifying text prompts. Generating adverse weather conditions (*e.g.*, foggy, night-time) particularly important to improve domain generalization for urban-scene segmentation, as described in DGInStyle (Jia et al., 2023). Therefore, we simply add a weather condition to the default prompt *e.g.*, "photorealistic first-person urban street view" to "... in foggy weather", as illustrated in Stage 4 in Fig. 2. Our Selective LoRA plays a critical role in generating diverse image-label pairs. As described in Fig. 1, fine-tuning a T2I model with original LoRA often causes overfitting to the undesired concept from the source dataset (*e.g.*, clear-day style from the Cityscapes (Cordts et al., 2016)). In contrast, fine-tuning only the viewpoint-sensitive weights can provide an exclusive learning viewpoint concept from the source dataset, which can effectively preserve the text adherence of the T2I model except for viewpoint (*e.g.*, do not memorize clear-day style). Additionally, regarding the in-domain scenario, we can generate diverse images by varying the class names used as arguments in the prompt template (*e.g.*, "... with car, person, etc.").   NEW

## 4 EXPERIMENTS

The following sections present extensive experiments to improve urban-scene segmentation in both in-domain and domain generalization (DG) settings. Section 4.1 describes the experimental setup and implementation details. Then, Section 4.2 presents extensive urban-scene segmentation experiments across in-domain and DG settings. Finally, Section 4.3 provides an in-depth analysis of the Selective LoRA, including a comprehensive ablation study.   NEW

### 4.1 EXPERIMENTAL SETUP

**Datasets**   For the training dataset, we utilize the Cityscapes (Cordts et al., 2016) as a source dataset for all urban-scene segmentation experiments, including in-domain and DG settings. Regarding the experiments for the in-domain few-shot setting, we only utilize a subset of Cityscapes images for the few-shot samples. For the evaluation, we test on Cityscapes validation set for in-domain, and ACDC (Sakaridis et al., 2021), Dark Zurich (DZ) (Sakaridis et al., 2019), BDD100K (BDD) (Yu et al., 2020), and Mapillary Vistas (MV) (Neuhold et al., 2017) for DG settings. Notably, ACDC and DZ are constructed with adverse weather conditions. We also conducted experiments with a general segmentation dataset using Pascal-VOC, which we included the results in Appendix A.8.   NEW

**In-Domain Semantic Segmentation**   For the baseline model, we train Mask2Former (Cheng et al., 2022) using subsets of the Cityscapes (Cordts et al., 2016) dataset at various fractions (0.3%, 1%, 3%, 10%). For the 100% (fully-supervised), we utilize the pretrained Mask2Former checkpoint. Then, we generated a total of 500 image-label pairs for all few-shot settings and used them as an   FIX

Table 1: In-domain segmentation performance across various fractions of the Cityscapes dataset (mIoU). In the first row, we trained Mask2Former on various fractions of the Cityscapes dataset (Baseline). Then, we fine-tuned the baseline on DatasetDM and our generated datasets with 30K iterations and evaluated the performance of the fine-tuned segmentation models. Additionally, we include an additional fine-tuned baseline (Baseline (FT)) that is solely fine-tuned on the same real dataset for a fair comparison in terms of the total iterations.

| Method | Training Dataset | | Total Iterations | Fraction of the Cityscapes Dataset | | | | |
|---|---|---|---|---|---|---|---|---|
| | Real | Generated | | 0.3% | 1% | 3% | 10% | 100% |
| Baseline | ✓ | ✗ | 90K | 41.83 | 49.15 | 59.07 | 69.02 | 79.40 |
| For a fair comparison, we fine-tune the baseline for additional 30K iterations using real or generated datasets. | | | | | | | | |
| Baseline (FT) | ✓ | ✗ | 120K | 42.00 (+0.17) | 49.18 (+0.03) | 59.06 (-0.01) | 68.68 (-0.34) | 80.05 (+0.65) |
| DatasetDM | ✓ | ✓ | 120K | 42.82 (+0.99) | 49.71 (+0.56) | 60.31 (+1.24) | 69.04 (+0.02) | 80.45 (+1.05) |
| Ours | ✓ | ✓ | 120K | **44.13 (+2.30)** | **51.90 (+2.75)** | **61.29 (+2.22)** | **70.29 (+1.27)** | **80.74 (+1.34)** |

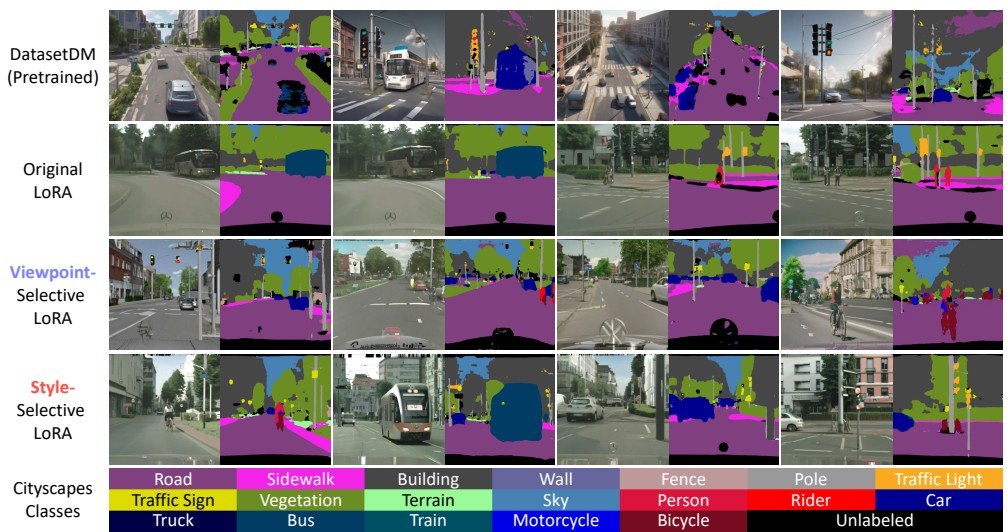

Figure 6: Qualitative comparison of image-label alignment between DatasetDM, Original LoRA, and our Viewpoint- and Style-Selective LoRA in a few-shot setting (Cityscapes 0.3%).

additional dataset to fine-tune the baseline model trained on each Cityscapes fraction, where we generated 3000 pairs for the fully-supervised setting. To avoid overfitting to the generated data, we mix real (*i.e.*, the dataset used during pretraining) and generated samples in each mini-batch in equal numbers. We have compared our proposed approach with the segmentation dataset generation approach, DatasetDM (Wu et al., 2023a), which utilizes the pre-trained text-to-image generation model (T2I model) without any fine-tuning. We also include an additional baseline (Baseline (FT)) FIX that fine-tunes the pretrained Mask2Former exclusively on the same real dataset, ensuring a fair comparison in terms of computational cost.

**Domain Generalization in Semantic Segmentation** We improve the DG performance for urban-scene segmentation upon the existing DG methods ColorAug, DAFormer (Hoyer et al., 2022a) and HRDA (Hoyer et al., 2022b), following the DGInStyle experimental setup (Jia et al., 2023). We have compared our proposed approach with the recent dataset generation approaches, DGIn-Style (Jia et al., 2023), DatasetDM (Wu et al., 2023a), DATUM (Benigmim et al., 2023), and NEW InstructPix2Pix (Brooks et al., 2023). [2] To show the effectiveness of the generated dataset, we train a semantic segmentation model with each DG method on a combination of the 2975 Cityscapes image-label pairs and the 2500 generated image-label pairs (500 images for 5 weather conditions including clear, foggy, night-time, rainy, and snowy) from scratch.

**Implementation Details** Throughout the experiments, we utilize Stable Diffusion XL (Podell et al., 2023) as the pretrained T2I model. The implementation is based on HuggingFace Diffusers library code (von Platen et al., 2022). We fix the rank of both the original LoRA and Selective LoRA at

---

[2]For DATUM, we provide an additional image for each weather condition to meet the requirements of its One-shot UDA setting. For InstructPix2Pix, we modify the weather conditions of the images using instruction prompts (e.g., "change the weather condition to {...}") while preserving the original segmentation maps.

Table 2: Comparison of generated datasets for domain generalization (DG) in urban-scene segmentation (Cityscapes → ACDC, Dark Zurich, BDD100K, Mapillary Vistas). The experiments are conducted upon various DG methods (ColorAug (Xie et al., 2021), DAFormer (Hoyer et al., 2022a), and HRDA (Hoyer et al., 2022b)). We employ the Viewpoint-Selective LoRA to generate our generated dataset. * The gray color indicates the reported score from the DGInStyle authors.[3] †DATUM additionally leverages a single target domain image for each weather condition from the ACDC dataset.

| DG Method | Generated Dataset | ACDC | DZ | BDD | MV | Average |
|---|---|---|---|---|---|---|
| ColorAug* | ✗ | 52.38 | 23.00 | 53.33 | **60.06** | 47.19 |
| ColorAug* | DGInStyle | **55.19** | **26.83** | **55.18** | 59.95 | 49.29 (+2.10) |
| ColorAug | ✗ | 53.12 | 25.69 | 53.00 | 59.81 | 47.91 |
| ColorAug | DatasetDM | 53.80 | 27.70 | 53.54 | 60.75 | 48.95 (+1.04) |
| ColorAug | InstructPix2Pix | 56.02 | 26.92 | 54.03 | 60.44 | 49.35 (+1.44) |
| ColorAug | Ours | **56.07** | **29.75** | **54.35** | **61.40** | **50.39 (+2.49)** |
| DAFormer* | ✗ | 55.15 | 28.28 | 54.19 | 61.67 | 49.82 |
| DAFormer* | DGInStyle | **57.74** | **28.55** | **56.26** | **62.67** | 51.31 (+1.48) |
| DAFormer | ✗ | 53.98 | 27.82 | 54.29 | 62.69 | 49.70 |
| DAFormer | DatasetDM | 55.24 | 28.44 | 54.40 | **63.18** | 50.32 (+0.62) |
| DAFormer | InstructPix2Pix | 55.13 | 26.93 | 54.61 | 62.36 | 49.76 (+0.06) |
| DAFormer† | DATUM | 54.06 | 27.10 | **54.74** | 62.40 | 49.58 (-0.12) |
| DAFormer | Ours | **55.83** | **31.68** | 54.68 | 63.09 | **51.32 (+1.63)** |
| HRDA* | ✗ | 59.70 | 31.07 | 58.49 | **68.32** | 54.40 |
| HRDA* | DGInStyle | **61.00** | **32.60** | **58.84** | 67.99 | 55.11 (+0.71) |
| HRDA | ✗ | 58.48 | 29.46 | 56.12 | 64.27 | 52.08 |
| HRDA | DatasetDM | 58.11 | 31.51 | 55.74 | 64.49 | 52.46 (+0.38) |
| HRDA | InstructPix2Pix | 58.50 | 29.56 | 56.10 | 64.10 | 52.07 (-0.01) |
| HRDA† | DATUM | 58.11 | 30.18 | **56.94** | 64.29 | 52.38 (+0.30) |
| HRDA | Ours | **58.93** | **34.41** | 56.56 | **64.54** | **53.61 (+1.53)** |

64 and set 10k training iterations for a fair comparison. While DatasetDM necessitates 20 hours for training the label generator in Stage 3, fine-tuning Selective LoRA only takes *one hour* on a single Tesla V100 GPU, which is a minimal amount of time compared to the entire training time. The selected proportion of Selective LoRA has been searched across 1%, 2%, 3%, 5%, and 10%. The diffusion timestep for identifying the desired concept is 81 across the 1000 timesteps. The results of our hyper-parameter search are reported in Appendix A.5. Additional implementation details, including the label generator architecture, hyper-parameters, number of generated pairs, and pseudo-code, are provided in Appendices A.1, A.2 and A.3.

FIX

## 4.2 MAIN RESULTS ON THE SEMANTIC SEGMENTATION BENCHMARKS

This section demonstrates the superiority of Selective LoRA in improving urban-scene segmentation performance through both quantitative and qualitative analyses. For the main results on the in-domain semantic segmentation benchmark, we use the Style-Selective LoRA with a 2% layer proportion, trained to adapt to the Cityscapes-style images by aligning image distributions. Conversely, for DG, we employ the Viewpoint-Selective LoRA with a 3% layer proportion, as it needs to produce not only Cityscapes-style images but also examples reflecting adverse weather conditions, such as foggy or night-time scenes, which the Style-Selective LoRA cannot handle.

FIX

**In-Domain Semantic Segmentation** As shown in Tab. 1, the proposed Selective LoRA consistently outperforms all other methods across various data ratios. Specifically, the proposed method improves 2.30 mIoU for the 0.3% data ratio, significantly surpassing DatasetDM, which achieves only a 0.99 mIoU increase. This demonstrates the efficiency of Selective LoRA, particularly in low-data regimes. Furthermore, our method improves the fully-supervised performance (*i.e.*, 100% data ratio) by 1.34 mIoU, further enhancing strong Mask2Former performance. The consistent gains across different data ratios highlight the robustness of Selective LoRA, making it a highly effective approach for urban-scene segmentation in both few-shot and fully-supervised scenarios. Additionally, we observe Selective LoRA achieves enhanced image-label alignment compared to the baseline methods, as shown in Fig. 6, and we quantitatively evaluate the image-label alignment in Appendix A.6 due to the page limit of the main paper. We made an in-depth analysis of image-label alignment, a crucial factor to confirm when performing segmentation dataset generation.

FIX

---

[3]Due to the absence of the source code with the generated datasets in Cityscapes as a source dataset, we cannot reproduce the reported results. However, we will update the table when the code is available.

Table 3: Image domain alignment between the generated and training real images using CMMD ($\downarrow$) (Jayasumana et al., 2024). The Style-Selective LoRA consistently shows better alignment than the Viewpoint-Selective LoRA across the various proportions of the selected layers.

| Desired Concept | Proportion of T2I Model Layers for Selective Fine-tuning | | | | | | |
|---|---|---|---|---|---|---|---|
| | 0% (Pretrained) | 1% | 2% | 3% | 5% | 10% | 100% (Original LoRA) |
| Style | 5.063 | 1.618 | 1.420 | 1.021 | 1.105 | 0.686 | **0.644** |
| Viewpoint | | 2.650 | 2.313 | 1.580 | 1.733 | 1.476 | |

Table 4: Fidelity of the augmented prompts for generating adverse weather conditions (*e.g.*, foggy, night-time, rainy, and snowy) measured using CLIP Score ($\uparrow$). The Viewpoint-Selective LoRA consistently outperforms across the various proportions of the selected layers.

| Desired Concept | Proportion of T2I Model Layers for Selective Fine-tuning | | | | | | |
|---|---|---|---|---|---|---|---|
| | 0% (Pretrained) | 1% | 2% | 3% | 5% | 10% | 100% (Original LoRA) |
| Style | 25.72 | 21.44 | 19.92 | 20.74 | 19.72 | 19.86 | 22.66 |
| Viewpoint | | 25.03 | 24.69 | **25.88** | 25.52 | 24.76 | |

**Domain Generalization in Semantic Segmentation** As shown in Tab. 2, the proposed method consistently outperforms all other segmentation dataset generation methods across multiple DG methods. Specifically, Viewpoint-Selective LoRA effectively learns only the viewpoint from the source dataset (Cityscapes) while maintaining the ability to generate diverse styles from the pretrained T2I model. As a result, our method significantly improves generalization performance, particularly on challenging datasets such as ACDC and Dark Zurich, where conditions such as adverse weather play a critical role. We emphasize that DGInStyle and InstructPix2Pix only change the styles of given images while keeping fixed label maps, which introduces limited manipulation of the scene. Since DAFormer and HRDA already employ strong image augmentation techniques (Hoyer et al., 2022a;b), the additional image augmentations from DGInStyle and InstructPix2Pix are largely redundant, as shown in their performance. In contrast, the improvement of the proposed approach is notable not only in comparison to the simple baseline (ColorAug), but also in its effectiveness on the advanced DG methods (DAFormer and HRDA), further proving the robustness of our approach.    NEW

## 4.3 ANALYSIS

This section presents an in-depth analysis of the Selective LoRA finetuned models (style and viewpoint), comparing them to the pretrained model and the original LoRA finetuned model. First, we evaluate the effective style adaptation of the Style-Selective LoRA. Next, we test the preservation of conditional image generation ability of the Viewpoint-Selective LoRA. Finally, we show a comprehensive ablation study for in-domain and DG for urban-scene segmentation across the pretrained model, Selective LoRA finetuned models (style, viewpoint), and the original LoRA finetuned model. Additional experimental studies of the concept sensitivity are available in Appendices A.7 and A.9.    NEW

**Image Domain Alignment** In this section, we evaluate the image domain alignment of the pretrained and finetuned T2I models, which is crucial for in-domain dataset generation. Since our analysis involves few-shot experiments (*e.g.*, 0.3%), we adopt CMMD (Jayasumana et al., 2024) for image alignment metric due to its consistent performance on small datasets. Tab. 3 shows that the pretrained T2I model exhibits a significant domain gap between real and generated images. In contrast, fine-tuning the T2I model on the Cityscapes dataset effectively reduces the domain gap. Among the Selective LoRAs, Style-Selective LoRA achieves competitive image domain alignment with only a 10% proportion compared to the original LoRA. While the original LoRA achieves the best alignment in the CMMD metric, Fig. 1 highlights its memorization problem, which we analyze further in the following ablation study to demonstrate the inferiority of the memorized dataset.

**Image Generation for Various Weather Scenarios** Since we generate diverse weather conditions (*e.g.*, foggy, night-time, rainy, and snowy) to improve the DG performance, preserving the conditional image generation ability is crucial. Thus, we measure the diverse weather conditional image generation performance by leveraging CLIP-Score (Radford et al., 2021), which can assess the similarity between the generated images and their input text prompts. We measure the average CLIP-Score across the four diverse weather conditions by generating 100 images for each weather condition. As shown in Tab. 4, the pretrained model shows a high CLIP score for generating adverse weather conditions, while the original LoRA cannot generate the weather conditions. Furthermore,

Table 5: Ablation study of the selected layers on the few-shot segmentation (Cityscapes 0.3% of labeled samples). The Style-Selective LoRA with a 2% proportion of the selected layers has shown the best performance.

| Desired Concept | Proportion of T2I Model Layers for Selective Fine-tuning | | | | | | |
|---|---|---|---|---|---|---|---|
| | 0% (Pretrained) | 1% | 2% | 3% | 5% | 10% | 100% (Original LoRA) |
| Style | 42.82 | 43.77 | **44.13** | 43.94 | 43.05 | 43.36 | 42.97 |
| Viewpoint | | 43.13 | 43.01 | 42.37 | 42.08 | 42.52 | |

Table 6: Ablation study of the Selective LoRA in domain generalization. We utilize color augmentation additional to our generated dataset when performing domain generalization using Cityscapes as the source domain and ACDC, Dark Zurich, BDD100K, and Mapillary Vistas as the target domains.

| DG Method | Additional Generated Dataset | ACDC | DZ | BDD | MV | Average |
|---|---|---|---|---|---|---|
| ColorAug | - | 53.12 | 25.69 | 53.00 | 59.81 | 47.91 |
| ColorAug | Pretrained (DatasetDM) | 53.80 | 27.70 | 53.54 | 60.75 | 48.95 (+1.04) |
| ColorAug | Original LoRA | 54.25 | 28.42 | 54.34 | 61.42 | 49.61 (+1.70) |
| ColorAug | Selective LoRA (Style) | 52.55 | 26.42 | 54.04 | **61.81** | 48.71 (+0.80) |
| ColorAug | Selective LoRA (Viewpoint) | **56.07** | **29.75** | **54.35** | 61.40 | **50.39** (+2.48) |

the Style-Selective LoRA scores even worse than the original LoRA since it aims to learn the style from the source dataset, which includes the source weather (*e.g.*, clear-day weather). In contrast, the Viewpoint-Selective LoRA effectively preserves the adverse weather conditional generation performance while learning the viewpoint from the source dataset.

**Ablation Study** We conduct the ablation study of the Selective LoRA on the few-shot segmentation (0.3% Cityscapes) and also show the hyperparameter impact across selected ratios (1%, 2%, 3%, 5%, and 10%) and desired concepts (style and viewpoint). As shown in Tab. 5, the pretrained model and original LoRA show poor performance due to domain misalignment and memorization, respectively. In contrast, the style-selective LoRA consistently improves the performance than viewpoint-selective LoRA, and the 2% selected layer proportion of the style-selective LoRA shows the best performance across the variants.

Furthermore, we conduct an ablation study of the Selective LoRA on the simple DG method (ColorAug), with the fixed 3% proportion of the selected layers of Selective LoRA. As shown in Tab. 6, Viewpoint-Selective LoRA shows significant improvements on average. While the original LoRA and style-selective LoRA show competitive performance improvements on BDD100K and Mapillary Vistas, viewpoint-selective LoRA has significantly improved ACDC and Dark Zurich, which contain images under challenging weather conditions. These results also show the strength of the viewpoint-selective LoRA in synthesizing adverse weather conditions.

## 5 CONCLUSION AND FUTURE WORK

This paper proposes Selective LoRA, a novel fine-tuning method designed to learn only the desired concepts (*e.g.*, viewpoint or style) from the training dataset to generate semantic segmentation datasets. Our method effectively identifies and updates only the weights relevant to the desired concepts, enabling the fine-tuned image generation model to produce well-aligned and informative samples. Although the additional information provided by the generated datasets is constrained by the pretrained T2I model, we demonstrated notable improvements in segmentation performance across various settings, including in-domain (few-shot and fully-supervised) and domain generalization tasks. Our fine-tuning method shows great potential for learning only the desired concepts from training data, even when it includes unnecessary concepts, contributing to the field of dataset generation. The following are potential future directions for our work. First, while our primary focus was on reducing domain shifts in a pretrained T2I model for urban scene segmentation, extending segmentation dataset generation to more general datasets (*e.g.*, Pascal-VOC (Everingham et al., 2010), COCO (Lin et al., 2014)) remains an important challenge, as briefly explored in Appendix A.8. Second, while our experiments with Selective LoRA focus on segmentation dataset generation, this approach also shows potential for extracting specific concepts beyond style and viewpoint for personalized image generation, presenting a promising direction for future research.

FIX

FIX

## REPRODUCIBILITY STATEMENT

To ensure the reproducibility of our method, we present the experimental setup for each problem setting in Section 4.3. Additionally, details on implementation and evaluation can be found in Appendix A.1. The pseudocode for the overall training and testing scheme is provided in Appendix A.2. Along with relevant references and publicly available code, we believe our paper offers sufficient information for reimplementation.

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

# A  APPENDIX

## A.1  IMPLEMENTATION DETAILS

**Hyperparameters (Tables 14, 15, 16 and 17)**  We provide all hyperparameters to support reproducibility. In the first stage, we fine-tune Stable Diffusion XL (Podell et al., 2023) using the HuggingFace Diffusers library (von Platen et al., 2022). The specific hyperparameters for fine-tuning Stable Diffusion XL on the Cityscapes dataset are listed in Tab. 14, while the training configurations for the label generator can be found in Tab. 15.

Next, we train segmentation models for both in-domain and domain generalization scenarios. The hyperparameters for in-domain fine-tuning are provided in Tab. 16, while those for domain generalization, based on the DGInStyle (Jia et al., 2023) method, are included in Tab. 17. We hope these provided hyperparameters will facilitate reproducibility.

**Label Generator Architecture (Fig. 7)**  We build the label generator based on the recent segmentation dataset generation framework, DatasetDM (Wu et al., 2023a). The label generator in DatasetDM, called P-Decoder, is derived from the Mask2Former (Cheng et al., 2022) decoder architecture. It takes intermediate features from the T2I model, including feature maps and cross-attention maps. The label generator then concatenates features of the same resolution and reduces the feature dimensions using predefined projection layers. The multi-resolution feature maps are passed through the pixel decoder and the transformer decoder sequentially, which outputs the segmentation predictions. Finally, we calculate the loss function of the label decoder, which mirrors that of Mask2Former, incorporating binary cross-entropy, dice loss, and classification loss. However, several modifications exist between the original DatasetDM P-Decoder and our label generator due to architectural differences between Stable Diffusion v1.5 (Rombach et al., 2022) and Stable Diffusion XL (Podell et al., 2023).  FIX

Since DatasetDM is built on top of Stable Diffusion v1.5 (Rombach et al., 2022), we simply adjust the feature dimensions in the projection layers to accommodate Stable Diffusion XL (Podell et al., 2023). The detailed label generator architecture is illustrated in Fig. 7. However, if all feature maps and cross-attention maps are used, the total number of channels increases significantly, leading to an unmanageable number of parameters in the projection layers during concatenation and projection. In summary, the feature maps are extracted from the *last feature block* at each resolution of the upsampling blocks, while cross-attention maps are sampled at equal intervals (every 7 blocks) from the total 36 up-sampling blocks (i.e., 1st, 8th, ... 29th, 36th).  NEW

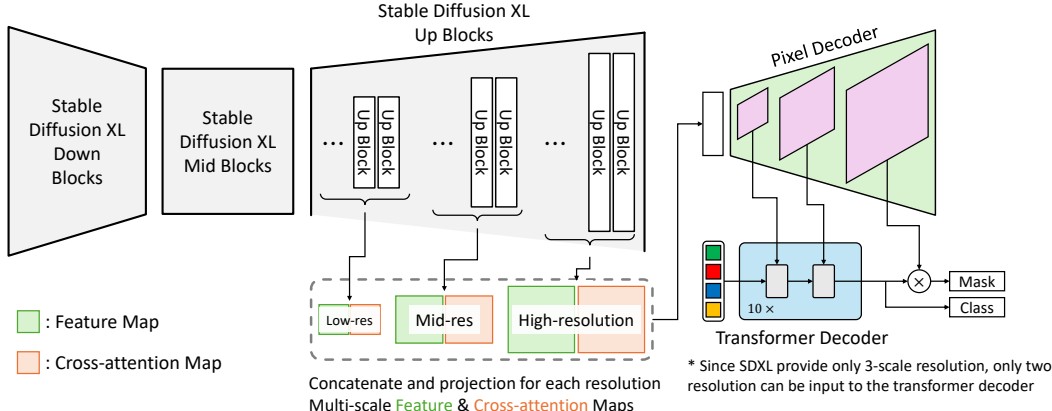

Figure 7: The detailed label generator architecture. The whole framework includes a text-to-image generation model (Stable Diffusion XL), pixel decoder, and transformer decoder, followed by DatasetDM (Wu et al., 2023a). Due to the change in the architecture of the text-to-image generation model, the following pixel decoder and transformer decoder minorly changed (*e.g.*, the number of input channels and the number of blocks).

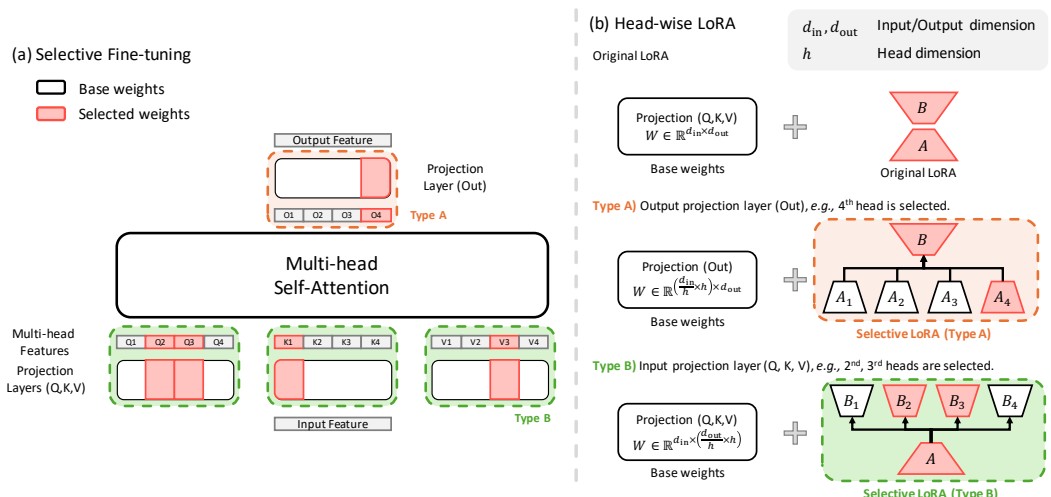

Figure 8: The detailed architecture of the Selective LoRA. We conduct head-wise Selective LoRA that can attach the LoRA layer for each head-wise projection layer.

Furthermore, as shown in Fig. 7, Stable Diffusion XL has only three resolution levels, compared to four resolution levels of the Stable Diffusion v1.5 architecture in the original DatasetDM. In the Mask2Former structure, feature maps from the pixel decoder, excluding the largest resolution, are fed into the transformer decoder. While the original design used three-resolution feature maps, only two were utilized in this case. Thus, while DatasetDM provides three-resolution feature maps three times for 9 transformer decoder blocks, we provide two-resolution feature maps five times, leading to a total of 10 transformer decoder blocks. *Importantly, to ensure a fair comparison, the reported scores for DatasetDM were obtained using a re-implemented version based on SDXL with the same modifications.*

NEW

## A.2 PSEUDOCODE

We also provide PyTorch-like pseudocode (Paszke et al., 2019) for key algorithms to effectively support reproducibility. **Concept sensitivity (Algorithms 1 and 2)** The concept sensitivity algorithm is demonstrated in Algorithm 1. Conducting concept sensitivity requires several helper functions,

as shown in Algorithm 2. **Selective LoRA (Algorithms 3 and 4)** The Selective LoRA algorithm is divided into two parts: the forward function and the declaration function. The forward pass of Selective LoRA is presented in Algorithm 3, while the declaration function, along with the selected layers, is illustrated in Algorithm 4.

While we provide the PyTorch-like pseudocode is based on HuggingFace Diffusers library (von Platen et al., 2022), HuggingFace PEFT-based implementation (Mangrulkar et al., 2022) can reduce the training time of the Selective LoRA.

### A.3 DETAILED ARCHITECTURE OF SELECTIVE LORA

**Overview (Fig. 8 (a))** The basic cluster of weights to measure the concept sensitivity is the projection layer. We selectively adapt the pretrained weights layer by layer within the projection layers. Since the LoRA layers are connected to the multi-head attention layers, the projection layers must be split head-wise to structurally distinguish their weights. Consequently, we also split the LoRA layers head-wise, as shown in Fig. 8.

**Head-wise Selective LoRA (Fig. 8 (b))** There are two types of Selective LoRA: output LoRA projection layers (Type A: Output (OUT)) and input LoRA projection layers (Type B: Query (Q), Key (K), Value (V)). To split the original LoRA layer ($\Delta W = BA$) in head-wise, the output projection LoRA layers ($\Delta W_{\text{OUT}}$) split the $A$ weights row-wise, while the input projection LoRA layers ($\Delta W_{\text{IN}}$) split the $B$ weights column-wise, as illustrated in the following equations.

$$\Delta W_{\text{OUT}} = B \begin{bmatrix} A_1 \\ A_2 \\ \vdots \\ A_h \end{bmatrix}, \qquad \Delta W_{\text{IN}} = \begin{bmatrix} B_1 & B_2 & \cdots & B_h \end{bmatrix} A. \tag{7}$$

The head-wise LoRA projection layer is represented in Fig. 8 and Algorithm 4.

### A.4 THE IMPLICIT BIAS OF GRADIENTS ACROSS THE LAYERS

**Observation (Fig. 9) (left)** We compute the sensitivity scores for each layer by using the norm of the gradient. However, the gradient norm cannot be uniformly scaled across different head types (Q, K, V, OUT), attention types (self, cross), and layers (shallow, deep). To address this, we construct a base gradient to scale the concept loss gradient by referencing the gradient of the original diffusion loss, as described in Section 3.2. We visualize the gradients of both the concept losses and the original diffusion loss in Fig. 9. In this visualization, we separate the self-attention and cross-attention layers to provide clearer distinctions, which differs from the approach in the main paper.

**Normalizing Gradients (Fig. 9) (right)** Therefore, we normalize the gradients using the gradients calculated from the original diffusion loss, as discussed in Section 3.2 and shown in Fig. 3. As shown in Fig. 9 (left), the gradients calculated from the style concept loss and viewpoint concept loss are similar. However, the gradient increase ratio can differ significantly, as illustrated in Fig. 9 (right).

### A.5 CONCEPT SENSITIVITY ACCORDING TO THE NOISE TIMESTEP

The amount of added noise (defined by the timestep, $t$) is the most crucial hyperparameter for concept sensitivity. We conduct extensive experiments to assess concept sensitivity in relation to the noise timestep. Visualizations of concept sensitivity across different noise timesteps, along with qualitative and quantitative results, are provided. These experimental results offer insights into the behavior of concept sensitivity.

**Visualization (Fig. 10)** According to our experiment, calculating concept sensitivity at large timesteps (noisy images) does not yield meaningful information about concept sensitivity. For example, style and viewpoint sensitivities appear similar when the timestep is set to 481 out of 1000, as shown in the first column of Fig. 10. This occurs because concept-sensitive layers are less responsive to noisy inputs, which have a high potential to generate any image. Conversely, extremely

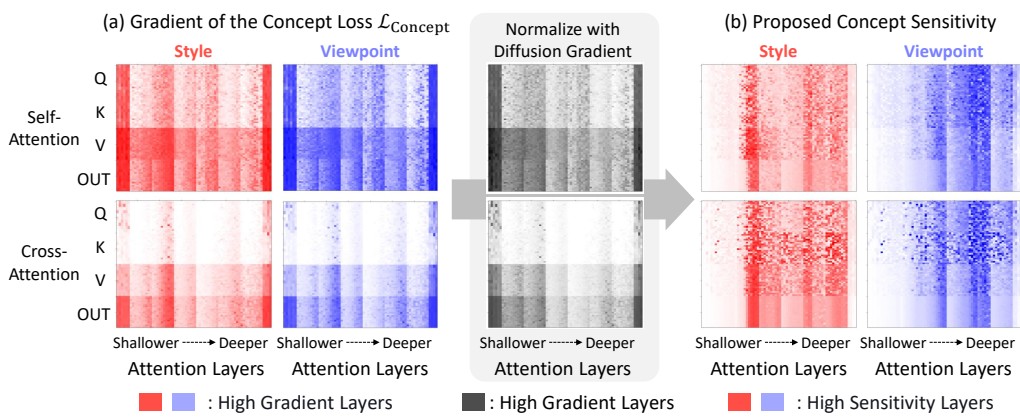

Figure 9: The implicit bias of the gradients across the layers (left). Concept sensitivity is calculated by normalizing each gradient with the gradient of the original diffusion loss (right).

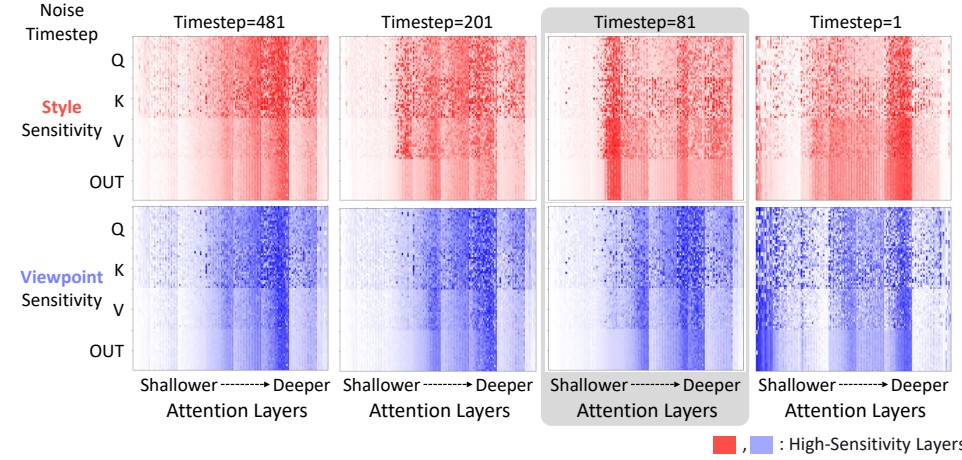

Figure 10: Visualizing concept sensitivity across different noise timesteps (481, 201, 81, and 1) shows that the 81st timestep stands out with a significantly distinct concept sensitivity score between style and viewpoint sensitivity compared to the other timesteps.

small timesteps (*e.g.*, 1) also fail to capture concept sensitivity, as the loss from almost clean images does not provide sufficient generative information. Therefore, we explored intermediate timesteps (*e.g.*, 201, 81) and found that the 81st timestep reveals distinct concept sensitivities for style and viewpoint.

**Qualitative Results (Fig. 11)** Additionally, we fine-tuned 2% of the selected ratio using each concept sensitivity and generated images to qualitatively compare results across different noise timesteps. As shown in Fig. 11, the images generated using intermediate timesteps (201, 81) better align with the intended style and viewpoint.

**Quantitative Results (Tab. 7)** Finally, we quantitatively compared the intermediate timesteps using the image domain alignment metric, CMMD ($\downarrow$) (Jayasumana et al., 2024), to evaluate the 201st and 81st timesteps. The results indicate that the 81st timestep is the most effective for measuring concept sensitivity, as shown in Tab. 7.

While our approach selects a single timestep to measure concept sensitivity, averaging multiple timesteps could improve the precision and robustness of concept sensitivity, which may be a promising direction for future research.

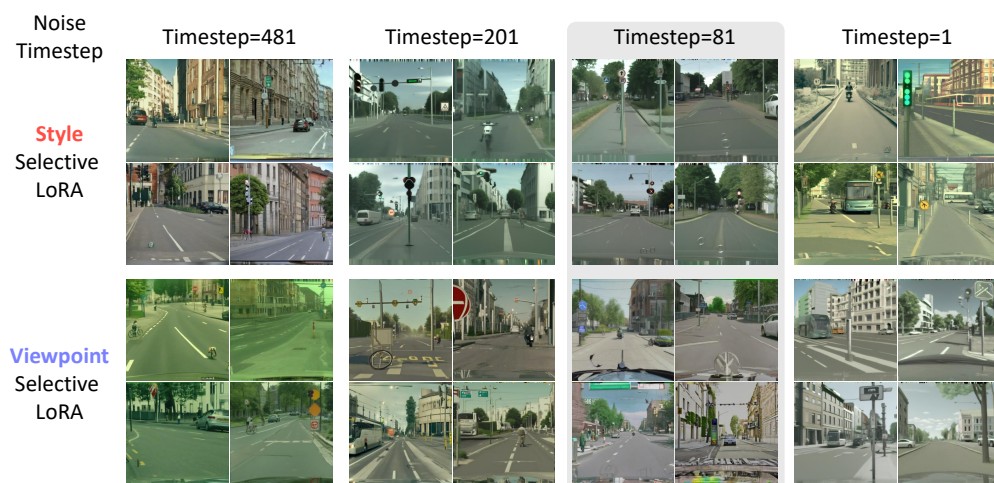

Figure 11: According to the various noise timestep, 81st timestep represents the best concept sensi-tivity, qualitatively. The style of the generated images by style-selective LoRA is well-aligned, while the generated images by viewpoint-selective LoRA contain diverse styles.

Table 7: The CMMD (↓) (Jayasumana et al., 2024) of the 2% concept-selective LoRA (style, view-point) is evaluated across the extracted timesteps.

| Extracted Timestep | 481 | 201 | 81 | 1 |
|---|---|---|---|---|
| Style | 1.920 | 1.556 | 1.420 | 2.383 |
| Viewpoint | 1.626 | 2.132 | 2.313 | 2.555 |

## A.6 COMPARISON OF IMAGE-LABEL ALIGNMENT

**Quantitative Comparison (Tab. 8)** Since the generated images lack ground-truth label maps, we measure image-label alignment using predictions from a pretrained segmentor. Specifically, we use the predictions from the pretrained Mask2Former model, which was fully supervised on the 100% Cityscapes dataset and achieves a 79.40 mIoU, as a proxy for the ground truth mask. Since this method is valid only when the pre-trained segmentor significantly outperforms the label generator, we conduct the image-label alignment experiment in a 0.3% few-shot setting.

**Analysis of the Qualitative Comparison (Fig. 6)** We compare not only image quality but also image-label alignment across DatasetDM (Wu et al., 2023a), original LoRA, and our Viewpoint- and Style-selective LoRA in the 0.3% few-shot segmentation setting. As shown in the generated labels in Fig. 6, DatasetDM fails to generate corresponding labels, while our Style-Selective LoRA generates high-quality corresponding labels. We suppose the reason is grounded by the domain gap between the pretrained T2I model (SDXL (Podell et al., 2023)) and the source dataset (Cityscapes (Cordts et al., 2016)). As mentioned Section 3.4, DatasetDM trains a label generator using the images of the source domain (*e.g.*, Cityscape images) without performing domain adaptation of the pre-trained T2I model. In other words, the domain gap exists between the label generator and the text-to-image model since the label generator is updated with the Cityscapes images while the original pretrained text-to-image model is not. Due to this domain gap, the intermediate features extracted from the original text-to-image model often fail to reflect the knowledge required for generating label maps of Cityscapes when used as input for the label generator. On the other hand, Style-Selective LoRA effectively adapts the T2I model to generate Cityscapes-style images. Therefore, Style-Selective LoRA can generate high-quality labels by reducing the domain gap between the intermediate features. However, although the image-label alignment has increased according to the increasing pro-portions of the selected layers, it does not always provide a better dataset, as shown in our ablation study Tab. 5 due to the image memorization problem.

NEW

Table 8: Image-Label Alignment (mIoU) (↑) across the segmentation dataset generation approaches in a few-shot setting (0.3%).

| Desired Concept | Proportion of T2I Model Layers for Selective Fine-tuning | | | | | | |
|---|---|---|---|---|---|---|---|
| | 0% (Pretrained) | 1% | 2% | 3% | 5% | 10% | 100% (Original LoRA) |
| Style | 25.18 | 37.01 | 39.37 | 40.94 | 47.84 | 48.86 | **60.10** |
| Viewpoint | | 33.27 | 29.42 | 31.44 | 34.94 | 38.28 | |

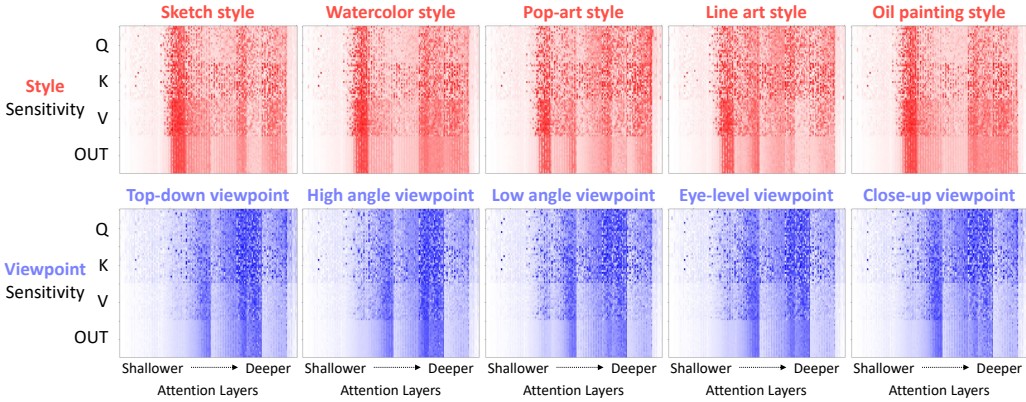

Figure 12: Measured concept sensitivity according to the various prompt augmentation. The highlighted concept-sensitive layers for each concept (style and viewpoint) remained largely consistent regardless of the prompt augmentation, demonstrating the robustness of concept sensitivity to variations in prompt design.

## A.7 CONCEPT SENSITIVITY ACCORDING TO THE PROMPT AUGMENTATION

In this section, we conduct an additional analysis of the robustness of defining desired concepts (Section 3.2) by showing measured sensitivity across the various prompt augmentations. Specifically, we provide five prompt augmentations from the original prompts, as shown in the following.    NEW

$$
c_{\text{Aug(Style)}} \in
\begin{cases}
\text{"Sketch of first-person urban street view",} \\
\text{"Watercolor of first-person urban street view",} \\
\text{"Pop-art of first-person urban street view",} \\
\text{"Line art of first-person urban street view",} \\
\text{"Oil painting of first-person urban street view"}
\end{cases}
\quad
c_{\text{Aug(Viewpoint)}} \in
\begin{cases}
\text{"Photorealistic urban street in top-down view",} \\
\text{"Photorealistic urban street in high angle view",} \\
\text{"Photorealistic urban street in low angle view",} \\
\text{"Photorealistic urban street in eye-level view",} \\
\text{"Photorealistic urban street in close-up view"}
\end{cases}
\quad (8)
$$

Then, we calculate the style and viewpoint sensitivity for each prompt augmentation, as shown in Fig. 12. As illustrated in the figure, our proposed method consistently demonstrates high sensitivity to similar regions across all prompt augmentations for styles. Similarly, for viewpoints, augmentations such as top-down, high-angle, and low-angle were applied, and the results indicate that our method highlights similar regions regardless of the specific viewpoint prompt. Based on these findings, we manually select the first three prompts for each desired concept. However, developing an automated approach to search for prompt augmentations could be a promising direction for enhancing concept sensitivity.    NEW

## A.8 IN-DOMAIN EXPERIMENTS FOR THE GENERAL DOMAIN DATASET (PASCAL-VOC)

Since our primary goal is to cover urban-scene segmentation, we focused on style and viewpoint as the desired concepts and conducted experiments exclusively on urban-scene datasets such as Cityscapes. However, the Selective LoRA methodology is not limited to urban-scene datasets. It can    NEW

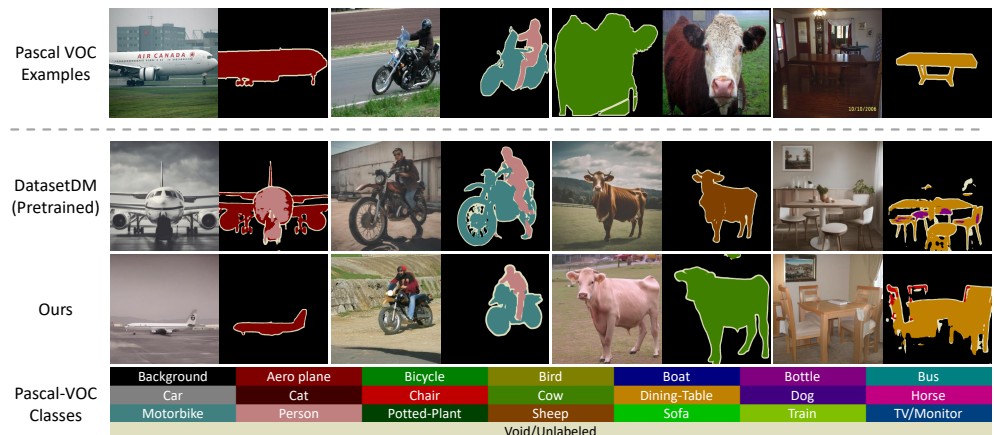

Figure 13: Qualitative comparison for generating Pascal-VOC dataset. Although both DatasetDM and ours are trained on the 100 labeled samples, our generated dataset shows better image domain alignment with the original Pascal-VOC examples and also shows better image-label alignment.

also be applied to general datasets for in-domain segmentation dataset generation. In this section, we demonstrate experiments on the Pascal-VOC dataset (Everingham et al., 2010), showcasing how our approach improves few-shot semantic segmentation performance.

**Experimental setup**     In this experiment, we trained on a total of 100 real image-label pairs and     NEW
evaluated the model using the 1,449 images in the Pascal-VOC validation set. For the text-to-image generation model, we applied the same style sensitivity score used in the Cityscapes experiment, setting the selected proportion to 10%. During the training of the text-to-image generation model, the prompt "a photo" was used. For training the label generator and generating the dataset, the prompt "a photo of a {class names}" was employed. The label generator was trained with a batch size of 4 for 90K iterations, ultimately producing 2,000 image-label pairs. When utilizing the generated dataset, the process was consistent with the in-domain semantic segmentation experiments. Specifically, Mask2Former was trained on the real dataset for 90K iterations (Baseline), followed by fine-tuning on the combined real and generated dataset for an additional 30K iterations. Additionally, we include an additional fine-tuned baseline (Baseline (FT)) that is solely fine-tuned on the same real dataset for a fair comparison in terms of the total iterations. All other hyper-parameters remained identical to those used in the Cityscapes in-domain semantic segmentation experiment, as detailed in Tables 14 to 16.

**Quantitative (Tab. 9) and Qualitative Results (Fig. 13)**     As shown in Tab. 9, using Style-     NEW
Selective LoRA on the Pascal-VOC dataset resulted in a performance improvement of 0.93 mIoU. In contrast, DatasetDM, which omitted the fine-tuning process for the text-to-image generation model, showed a performance drop of 8.43 mIoU. This highlights the importance of selective fine-tuning for style, even in general datasets beyond urban-scene datasets. Fig. 13 provides further insight into the role of style information. A significant image domain gap is evident between the images generated by the pretrained text-to-image generation model and the dataset generated using Pascal-VOC. This demonstrates the impact of image domain alignment. Quantitatively, the CMMD, which was 1.46 for the pretrained model, decreased to 0.81 after alignment, illustrating the reduced domain gap and its contribution to performance improvement.

A.9   COMPARISON WITH HAND-CRAFTED LAYER SELECTION APPROACHES

In this section, we aim to evaluate how effectively our sensitive weights identification method captures the desired concepts by comparing its performance with hand-crafted selected layers. This comparison is conducted by observing the improvement in in-domain semantic segmentation performance.

Table 9: In-domain segmentation performance (mIoU) of the Pascal-VOC dataset in the few-shot setting (100 image-label pairs). In the first row, we trained Mask2Former on various fractions of the Cityscapes dataset (Baseline). Then, we fine-tuned the baseline on DatasetDM and our generated datasets with 30K iterations and evaluated the performance of the fine-tuned segmentation models. Additionally, we include an additional fine-tuned baseline (Baseline (FT)) that is solely fine-tuned on the same real dataset for a fair comparison in terms of the total iterations.

| Method | Training Dataset | | Total Iterations | Segmentation Performance (mIoU) |
|---|---|---|---|---|
| | # Real | # Generated | | |
| Baseline | 100 | ✗ | 90K | 44.59 |
| For a fair comparison, we fine-tune the baseline for 30K iterations using the following datasets. | | | | |
| Baseline (FT) | 100 | ✗ | 120K | 44.39 (-0.20) |
| DatasetDM | 100 | 2,000 | 120K | 36.16 (-8.43) |
| Ours | 100 | 2,000 | 120K | **45.52 (+0.93)** |

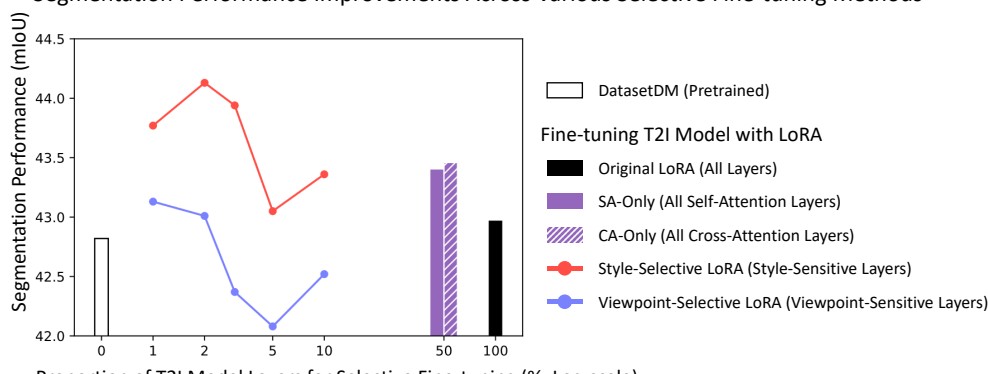

Figure 14: In-domain few-shot semantic segmentation comparison (0.3% Cityscapes) with the hand-crafted layer selection approaches. SA-Only and CA-Only indicate Selective LoRA fine-tuning approaches for all self- and cross-attention layers, respectively.

**Experimental setup**    The experimental setup is identical to the in-domain semantic segmentation experiment, presented in Section 4.2. We aim to improve performance by generating a segmentation dataset using 0.3% of the labeled Cityscapes dataset. For the hand-crafted manual selection baselines, we include "SA-Only," which fine-tunes only the self-attention layers with LoRA, and "CA-Only," which fine-tunes only the cross-attention layers with LoRA. To ensure a comprehensive comparison, we also evaluate the performance of "DatasetDM," which uses the pretrained model without fine-tuning, and "Original LoRA," which applies LoRA fine-tuning to all attention layers. Since the Stable Diffusion XL has self-attention and cross-attention layers equally, each hand-crafted layer selection method fine-tunes 50% of the total layers.    NEW

**Quantitative (Fig. 14) and Qualitative Result (Fig. 15)**    As shown in Fig. 14, SA-Only and CA-Only methods outperform DatasetDM and Original LoRA. However, their performance does not reach the level of our Style-Selective LoRA, which specifically targets the Style-Sensitive Layers in Cityscapes. To analyze this, we provide examples of bus samples generated using the prompt "photorealistic first-person urban street view *with bus*." for SA-Only, CA-Only, Original LoRA, and our Style-Selective LoRA. As illustrated in Fig. 15, the images of buses seen during training are limited to the top two examples. In the case of SA-Only, CA-Only, and Original LoRA, the generated buses closely resemble those seen during training, showing minimal variation. In contrast, our Style-Selective LoRA, which selectively fine-tunes only the Style-Sensitive Layers, is capable of generating a diverse range of buses while maintaining the Cityscapes style. We suppose that the diversity in the generated dataset of our method significantly contributed to the superior final performance improvements in semantic segmentation.    NEW

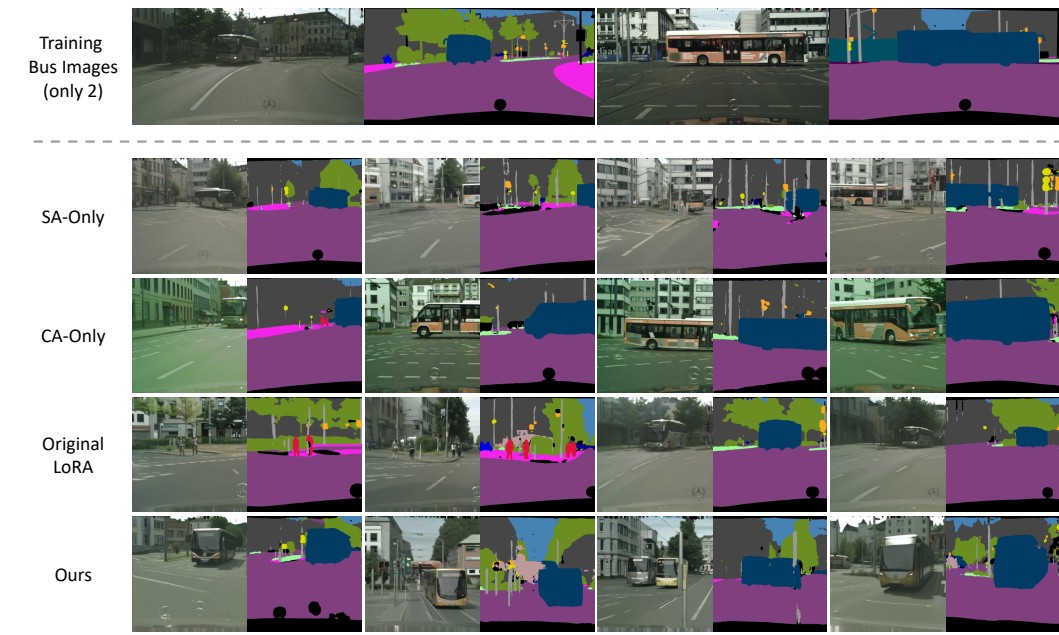

Figure 15: Qualitative comparison with the hand-crafted layer selection approaches by generating "bus" class. While SA-Only, CA-Only, and Original LoRA generate similar bus images with the training bus images, we can generate diverse bus images with well-aligned label maps.

## A.10 CLASS-WISE SEGMENTATION PERFORMANCE ANALYSIS

In this section, we present a detailed analysis of class-wise improvements, highlighting the effectiveness of the proposed method, particularly for rare classes. Additionally, we introduce a class-balanced performance improvement strategy tailored to specific classes.

**Class-wise Performance Improvements (Fig. 16)** Urban-scene segmentation has distinct challenges, including class imbalance and co-occurrence issues (Kim et al., 2024), making class-wise analysis particularly important. We present the class-wise IoU improvements in Fig. 16. As shown in Fig. 16 (a), our proposed dataset generation approach proves especially effective for rare classes such as "bus", "fence", and "bicycle". However, the generated dataset often fails to improve performance in certain classes, such as "person" and "rider". As illustrated in Fig. 16 (b), this degradation is primarily due to the insufficient number of generated samples for the "person" class. Since the synthetic dataset is generated randomly, disparities in label proportions can occur. To mitigate this, we propose a simple yet effective technique to increase the proportion of the target class.

**Segmentation Dataset Generation Focused on a Specific Class (Figures 16 and 17)** As detailed in Section 3.4 and Tab. 14, we generated the dataset using the prompt "photorealistic first-person urban street view with [Class names]", where the class names were extracted from the label map of the training set by retrieving the names of all classes present in the label map. While the synthesized text prompt partially reflects the label proportions of the training set, it does not strictly enforce these proportions. As a result, the proposed generated dataset may exhibit misaligned label proportions, as illustrated in Fig. 16 (b).

To address this issue, we propose a class-specific generation approach that manually increases the target class by modifying the generation prompts. Specifically, we generated an additional 500 samples using the prompt "photorealistic first-person urban street view with people" to increase the proportion of the "person" class.[4] Since we selectively fine-tuned the LoRA to learn only the style from Cityscapes, it enables effective manipulation using the text prompt, which the original

---

[4] We also experimented with "photorealistic first-person urban street view with person", but using "people" as the test prompt proved to be more effective in increasing the label proportion for the "person" class.

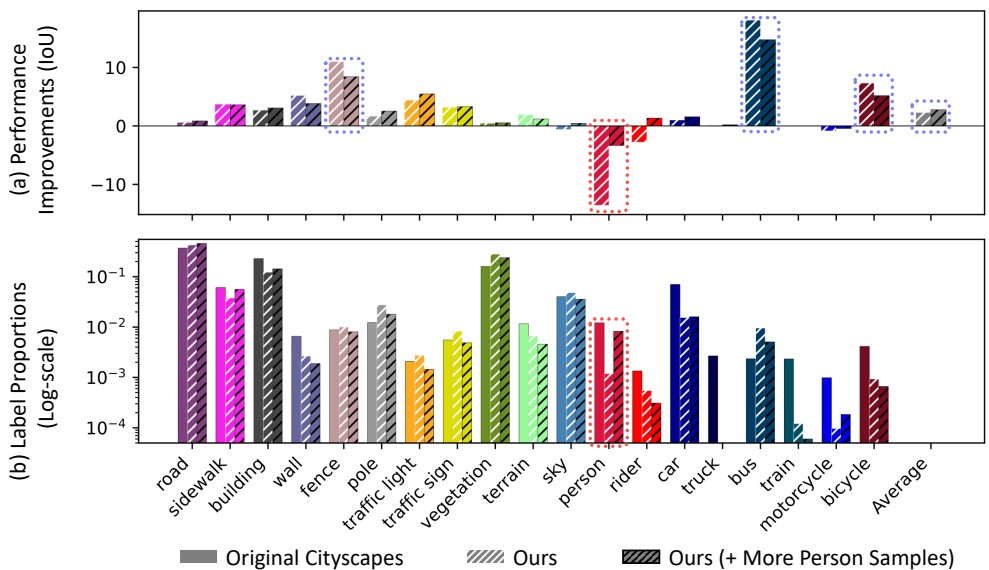

Figure 16: (a) Class-wise performance improvements (IoU) and (b) Label proportions for the original Cityscapes, Ours, and "Ours (+ More Person Samples)". "Ours (+ More Person Samples)" includes an additional 500 samples for the "person" class to balance the label proportions. (The additional baseline, "Ours", trained with the same number of images to match the size of the generated dataset, will be updated in the camera-ready version.) As shown in the class-wise performance, significant improvements were achieved for rare classes such as "bus", "fence", and "bicycle", as highlighted by the blue dotted lines. While some classes, such as "person" and "rider", showed degradation (indicated by red dotted lines), this was due to the lower number of generated samples for these classes. By generating additional samples for these specific classes, a more balanced performance improvement can be achieved, ultimately increasing the overall average performance.

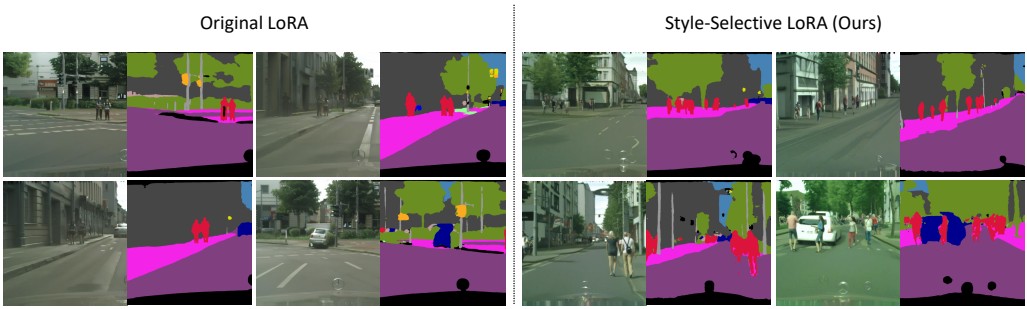

"Photorealistic first-person urban street view with people"

Figure 17: Qualitative comparison between the original LoRA and our Style-Selective LoRA for generating the "person" class to increase the label proportion. While the original LoRA can generate the "person" class, it is limited in producing informative samples beyond the training set, with generated images often resembling those from the training set. In contrast, the Style-Selective LoRA generates diverse scenes for the "person" class, as it exclusively learns the style from the source dataset.

LoRA cannot achieve, as demonstrated in Fig. 17. As illustrated in Fig. 16 (b), this approach successfully increased the proportion of the "person" class and mitigated its performance degradation. Furthermore, as shown in Fig. 16 (a), this adjustment led to additional performance improvements, increasing the average IoU from 44.12 to 44.59.

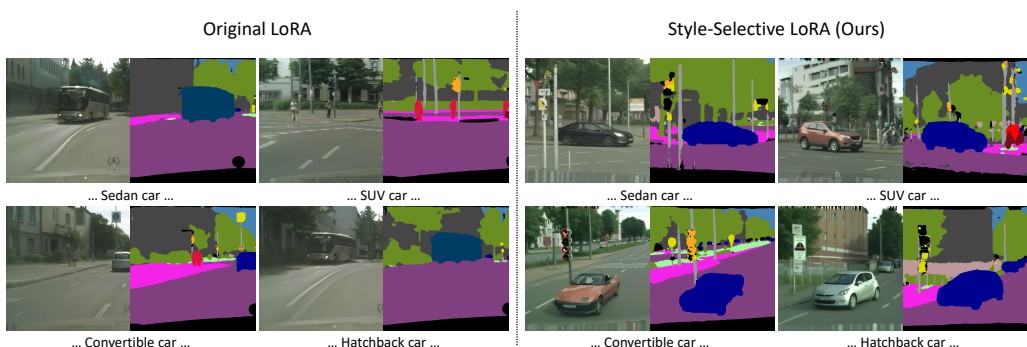

Figure 18: Generated image-label pairs showcasing various styles of cars, including sedan, SUV, convertible, and hatchback. Unlike the Original LoRA, since the Style-Selective LoRA exclusively learned only the style from the Cityscapes, we can generate various types of cars in Cityscapes-style.

Table 10: In-domain segmentation performance of datasets incorporating the diverse cars dataset. Incorporating the diverse cars dataset especially improved performance for vehicle classes such as "car", "bus", and "motorcycle", leading to overall performance improvements. Since we generated an additional 400 image-label pairs (100 images per vehicle type), the total number of the generated samples is 900. (The additional baseline, "Ours", trained with the same number of images to match the size of the generated dataset, will be updated in the camera-ready version.)

| Method | Training Dataset | | Total Iterations | IoU | | | mIoU |
|---|---|---|---|---|---|---|---|
| | # Real | # Generated | | Car | Bus | Motorcycle | |
| Baseline | 9 | ✗ | 90K | 84.02 | 12.51 | 16.11 | 41.83 |
| For a fair comparison, we fine-tune the baseline for 30K iterations using the following datasets. | | | | | | | |
| Baseline (FT) | 9 | ✗ | 120K | 83.98 | 13.37 | 15.96 | 42.00 |
| Ours | 9 | 500 | 120K | 85.03 | 30.59 | 15.26 | 44.12 |
| Ours (+ Diverse Cars) | 9 | 900 | 120K | **85.34** | **32.48** | **17.77** | **44.95** |

## A.11 GENERATING DATASETS WITH DIVERSE CLASS NAMES

Since the Style-Selective LoRA selectively fine-tuned only the style from the in-domain Cityscapes dataset, it retains its generalization ability for text prompts such as objects. Leveraging this capability, we aim to generate a broader variety of images using more diverse class names beyond those provided in the dataset. In this experiment, we refined the prompts for generating images previously created with the simple class name "car" by subdividing them into "sedan car", "SUV car", "convertible car", and "hatchback car", as shown in Fig. 18. As illustrated in the figure, while the original LoRA fine-tuned text-to-image generation model struggles to produce diverse styles of cars, our approach reliably generates a wide variety of cars that align with the test prompts.

NEW

We then conducted an in-domain few-shot experiment (Cityscapes 0.3%) using the additional diverse cars dataset, following the experimental setup described in Section 4.1. As shown in Tab. 10, incorporating the diverse cars dataset significantly improves segmentation performance, particularly for vehicle classes. Beyond generating diverse cars, applying textual augmentations to other class names for dataset creation represents a promising direction for advancing segmentation dataset generation.

NEW

## A.12 ADDITIONAL ANALYSIS OF OUR GENERATED DATASET ON THE DOMAIN GENERALIZATION SETTING

In this section, we aim to compare and analyze the performance of the Viewpoint-Selective LoRA against other baselines that have been applied to urban-scene segmentation in domain generalization. This analysis comprises qualitative assessments (Fig. 19) alongside quantitative evaluations of image domain alignment (Tab. 11) and image-label alignment (Tab. 13), similar to Section 4.3 and Appendix A.6, respectively.

NEW

Table 11: Comparison of image domain alignment with image generation baselines on four adverse weather conditions. The alignment is measured between the generated images and the ACDC dataset for each weather condition (CMMD ↓). †DATUM trained 4 models for each weather condition in the ACDC dataset, using an additionally provided single target domain image per condition.

| Method | Foggy | Night-time | Rainy | Snowy | Average |
|---|---|---|---|---|---|
| DATUM† | 2.41 | 2.46 | 2.91 | 2.10 | 2.47 |
| InstructPix2Pix | 3.43 | 3.13 | 2.99 | 3.32 | 3.22 |
| DatasetDM | 4.90 | 5.52 | 5.34 | 4.96 | 5.18 |
| Ours | 2.43 | 2.55 | 2.62 | 2.63 | 2.56 |

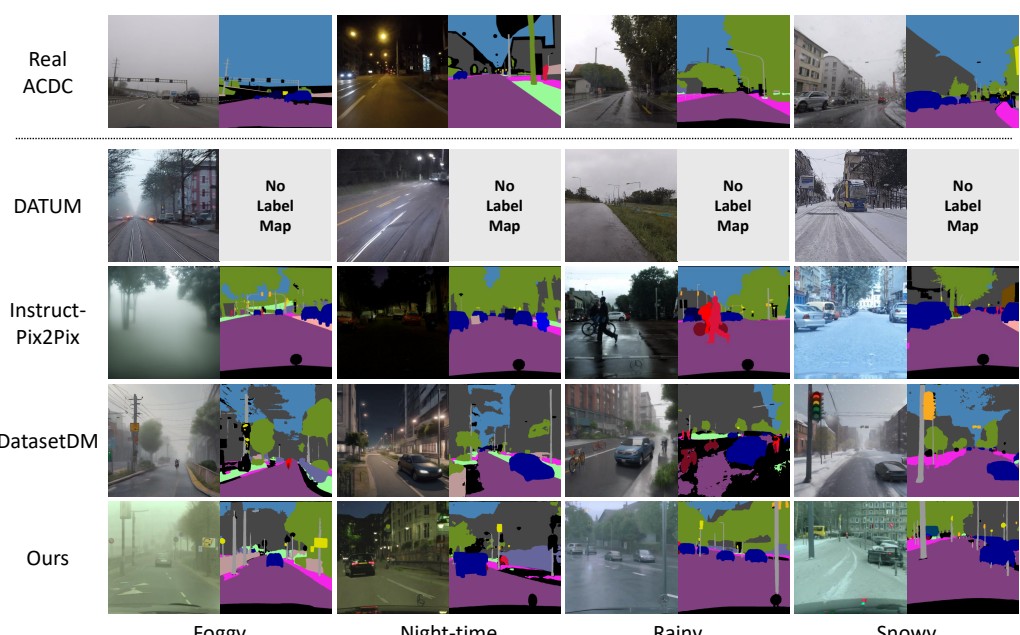

Figure 19: Qualitative results for generating image-label pairs in domain generalization settings. The proposed approach demonstrates its efficacy in both image domain alignment and image-label alignment.

**Image Domain Alignment (Tab. 11 and fig. 19)** For domain generalization in urban-scene segmentation, we generated urban-scene images under various adverse weather conditions (*e.g.*, "foggy", "night-time", "rainy", and "snowy"). In this section, we assess the domain gap between our generated adverse weather conditions and the real ACDC (Sakaridis et al., 2021) dataset. Qualitatively, as shown in Fig. 19, our approach generates images that are more realistic and better aligned compared to DatasetDM and InstructPix2Pix, which rely on pretrained models without fine-tuning. When compared to DATUM (Benigmim et al., 2023), our method achieves a similar level of image domain alignment while generating more diverse scenes. Quantitatively, we used CMMD (Jaya-sumana et al., 2024) to measure image domain alignment, and the results are presented in Tab. 11. These results show that the proposed generated dataset demonstrates a significant performance gap over DatasetDM and InstructPix2Pix. More importantly, it achieves competitive performance with DATUM, which requires training *individual models* separately for each weather condition *using a target domain image from the ACDC dataset*.

**Image-Label Alignment (Fig. 19 and tables 12 and 13)** We compare image-label alignment to evaluate how reliably label maps are generated for datasets aimed at domain generalization. Qualitatively, as shown in Fig. 19, DATUM generates only images and sets them as pseudo-target domains to apply UDA methods, meaning that no labels are generated. In the case of InstructPix2Pix, style transfer is performed on Cityscapes image-label pairs, using the labels from Cityscapes directly. While this ensures high-quality labels, severe editing can occasionally cause alignment issues, as

Table 12: Segmentation performance of the Pretrained and Finetuned Mask2Former (M2F) (Cheng et al., 2022) on the adverse weather condition dataset (ACDC (Sakaridis et al., 2021)). Starting with the pretrained M2F model trained on the Cityscapes dataset (Cordts et al., 2016), we further fine-tuned the model on the ACDC training set for each individual weather condition (learning rate is 3e-6, the batch size is 2, and the number of iterations is 30K). This approach resulted in highly effective segmentation models tailored to specific weather conditions, serving as pseudo ground-truth masks for evaluating image-label alignment in domain generalization settings.

| Method | Foggy | Night-time | Rainy | Snowy | Average |
|---|---|---|---|---|---|
| Pretrained M2F | 67.66 | 23.17 | 51.94 | 47.55 | 47.58 |
| Finetuned M2F | 78.54 | 52.16 | 66.23 | 74.79 | 67.93 |

Table 13: Comparison of image-label alignment with baselines. While InstructPix2Pix provides reliable image-label alignment by fixing Cityscapes labels and applying style transfer only to the weather conditions of the images, its ability to generate diverse scenes is constrained by the fixed labels. In contrast, when comparing methods that generate labels, our approach demonstrates better image-label alignment than DatasetDM.

| Method | Foggy | Night-time | Rainy | Snowy | Average |
|---|---|---|---|---|---|
| InstructPix2Pix | 25.98 | 48.60 | 63.04 | 40.66 | 44.57 |
| DatasetDM | 40.84 | 35.90 | 47.43 | 44.02 | 42.05 |
| Ours | 41.55 | 43.07 | 48.69 | 39.47 | 43.20 |

seen in the foggy examples. Finally, comparing DatasetDM and our method, both of which generate labels directly, shows that our approach achieves significantly better label generation quality compared to DatasetDM.

We then proceed to evaluate image-label alignment quantitatively. As discussed in Appendix A.6, the generated images lack actual ground truth for domain generalization datasets. Therefore, we rely on pseudo ground truth generated by a highly accurate segmentor. Since no off-the-shelf urban-scene semantic segmentation model consistently performs well across diverse domains, we took several steps to develop a more reliable segmentor. First, as described in Appendix A.6, we began with a pretrained Mask2Former (M2F) model trained on the full Cityscapes dataset. However, as shown in Tab. 12, this model is susceptible to adverse weather conditions. To address this limitation, we fine-tuned the pretrained M2F model individually for each of the four adverse weather conditions in the ACDC training set. Since this dataset is not accessible to DatasetDM or our method, the fine-tuned M2F models are guaranteed to outperform those methods. The specific performance improvements on the ACDC validation set are detailed in Tab. 12.   NEW

The results of measuring image-label alignment using the fine-tuned M2F models are presented in Tab. 13. As shown in the table, InstructPix2Pix, which directly uses Cityscapes labels and only slightly edits the weather conditions of the images, demonstrates an advantage in image-label alignment. Despite its high image-label alignment, we highlighted the limited performance improvements of InstructPix2Pix in Tab. 2 and Section 4.2, attributing this to the lack of scene diversity caused by its reliance on fixed segmentation label maps. When comparing methods that generate labels, our approach achieves better image-label alignment than DatasetDM. This improvement stems from our text-to-image generation model learning viewpoints from Cityscapes. As a result, even with the same label generator architecture, our finetuned text-to-image generation model provides representations with a smaller domain gap when generating images based on the Cityscapes dataset.   NEW

## A.13 ADDITIONAL QUALITATIVE RESULTS

**Additional examples (Figures 20, 21 and 22)**   We illustrate the changes in generated images using Style- and Viewpoint-Selective LoRA as the proportion of selected layers varies (1%, 2%, 3%, 5%, and 10%). As shown in Fig. 20, both Selective LoRAs effectively focus on the target concept with a small proportion of selected layers. However, as the proportion increases, other concepts are gradually learned, as demonstrated in the 10% layer selection. For example, the Viewpoint-Selective LoRA shows a slight adaptation to the Cityscapes style. This flexibility allows for manual   NEW

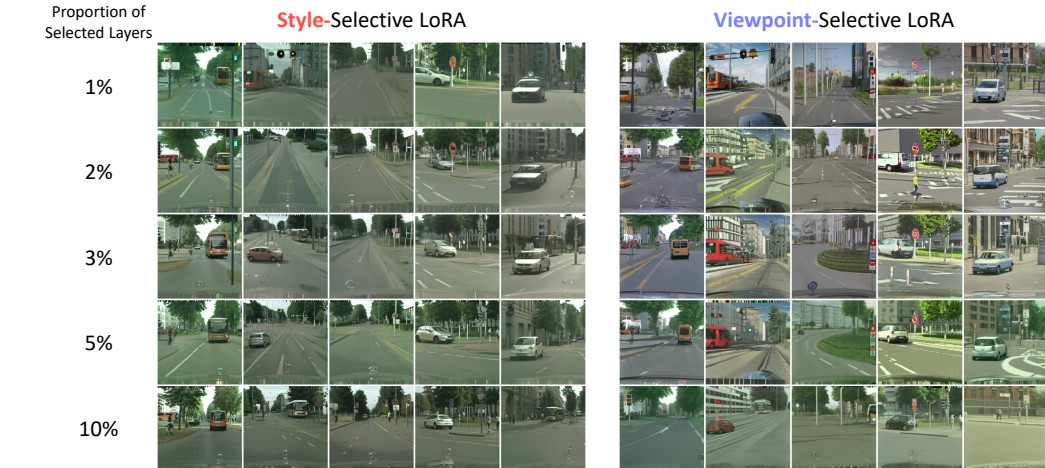

Figure 20: Qualitative results of Style- and Viewpoint-Selective LoRA according to the layer proportions (1%, 2%, 3%, 5%, and 10%). While the Style and Viewpoint-Selective LoRA effectively disentangle with the small proportions of the selected layers, it has been entangled according to the increased proportion of the selected layers. This flexibility allows for manual adjustment of the extent to which other concepts are learned, depending on the specific problem settings or datasets.

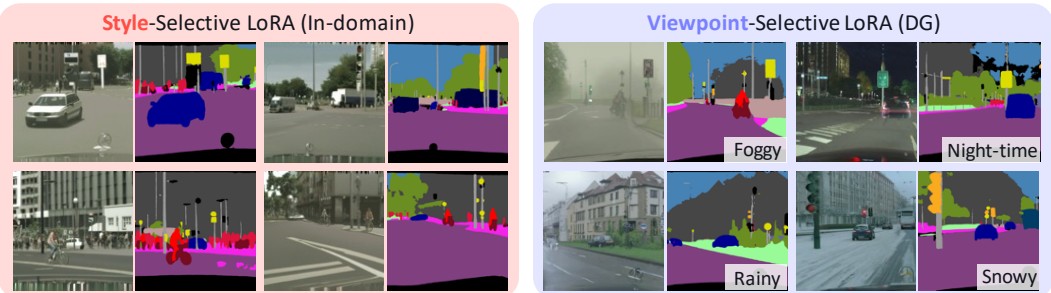

Figure 21: Qualitative examples of the generated image-label pairs for in-domain and domain generalization settings. Style-Selective LoRA effectively generates a Cityscapes-style dataset. Viewpoint-Selective LoRA can control the weather condition of the generated images with corresponding label maps since it selectively learns the Cityscapes-viewpoint.

adjustment of the extent to which other concepts are learned, depending on the specific problem settings or datasets.

Furthermore, we provide additional examples of the generated datasets used to improve segmentation performance in fully supervised and domain generalization settings. The additional examples of our generated datasets are available in Figures 21 and 22.

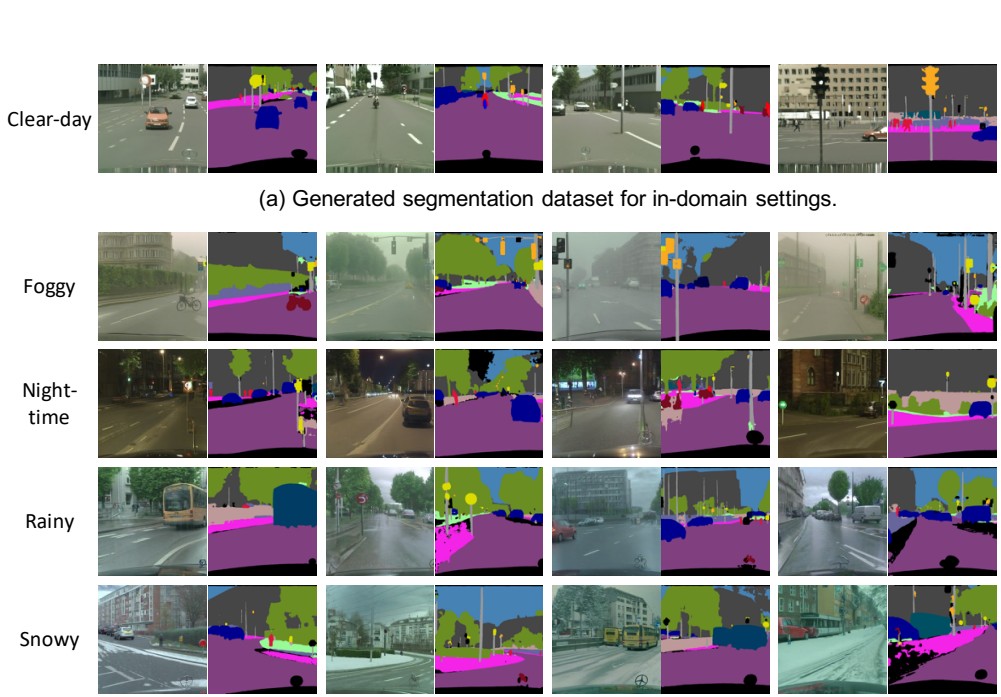

Clear-day

(a) Generated segmentation dataset for in-domain settings.

Foggy

Night-time

Rainy

Snowy

(b) Generated segmentation datasets for domain generalization.

Figure 22: Additional examples of our generated datasets for fully supervised and domain generalization scenarios.

Table 14: Hyperparameters to fine-tune Stable Diffusion XL (Podell et al., 2023). The class names are extracted from the label map in the training set by retrieving the names of all classes that appear in the label map.

| Hyperparameter | Value |
|---|---|
| Rank | 64 |
| Learning rate | 1e-4 |
| Batch size | 1 |
| Training iteration | 10K |
| Data augmentation | Random horizontal flip, Random crop |
| Resolution | (1024, 1024) |
| Learning rate scheduler | constant |
| Optimizer | AdamW (Loshchilov & Hutter, 2019) |
| Adam beta1 | 0.9 |
| Adam beta2 | 0.999 |
| Adam weight decay | 0.01 |
| Training prompt | "photorealistic first-person urban street view" |
| Test-time hyperparameters | |
| Num. inference steps | 25 |
| Guidance scale | 5.0 |
| Test prompt augmentation (In-domain) | "... with [Class names]"[5] |
| Test prompt augmentation (DG) | "... in [Weather Condition] with [Class names]" |

Table 15: Hyperparameters to train label generator followed by DatasetDM (Wu et al., 2023a).

| Hyperparameter | Value |
|---|---|
| Architecture | Mask2Former-shaped label generator (Wu et al., 2023a) |
| Learning rate | 1e-4 |
| Batch size | 2 for all few-shot, 8 for fully-supervised |
| Training iteration (few-shot) | 12k, 24k, 24k, and 48k for 0.3%, 1%, 3%, and 10%, respectively |
| Training iteration (fully-supervised) | 90K |
| Data augmentation | Random horizontal flip, Random resized crop (0.5, 2.0) |
| Resolution | (1024, 1024) |
| Learning rate scheduler | PolynomialLR(power=0.9) |
| Optimizer | Adam (Kingma, 2014) |
| Adam beta1 | 0.9 |
| Adam beta2 | 0.999 |
| Adam weight decay | 0.0 |

Table 16: Hyperparameters to fine-tune Mask2Former (Cheng et al., 2022). We modify the learning rate, batch size and training iteration from the original Mask2Former training configuration.

| Hyperparameter | Value |
|---|---|
| Model Architecture | Mask2Former (Cheng et al., 2022) |
| Num. generated images | 500 for all few-shot, and 3,000 for fully-supervised |
| Learning rate | 3e-6 |
| Batch size | 2 for all few-shot, and 8 for fully-supervised |
| Mixed batch | real:syn = 1:1 |
| Training iteration | 30K |
| Data augmentation | Random horizontal flip, Random resized crop (0.5, 2.0) |
| Resolution | (512, 1024) |
| Learning rate scheduler | PolynomialLR(power=0.9) |
| Optimizer | AdamW (Loshchilov & Hutter, 2019) |
| Adam beta1 | 0.9 |
| Adam beta2 | 0.999 |
| Adam weight decay | 0.05 |

Table 17: Hyperparameters to train domain generalization in segmentation including ColorAug, DAFormer (Hoyer et al., 2022a), and HRDA (Hoyer et al., 2022b), followed by DGInStyle (Jia et al., 2023).

| Hyperparameter | Value (ColorAug) (Xie et al., 2021) | Value (DAFormer) (Hoyer et al., 2022a) | Value (HRDA) (Hoyer et al., 2022b) |
|---|---|---|---|
| Model Architecture | SegFormer | DAFormer | HRDA |
| Backbone | | MiT-B5 (Xie et al., 2021) | |
| Num. generated images | | 500 for each weather condition (clear, foggy, night-time, rainy, and snowy) | |
| Learning rate | | 6e-5 | |
| Batch size | | 2 | |
| Training iteration | | 40K | |
| Data augmentation for Gen. | | Random horizontal flip, PhotoMetricDistortion | |
| Data augmentation for Real | | Random horizontal flip, Random crop, DACS (Tranheden et al., 2021) | |
| Resolution | (512, 512) | (512, 512) | (1024, 1024) |
| Learning rate scheduler | | PolynomialLR(power=0.9) | |
| Learning rate warmup | | Linear | |
| Learning rate warmup iteration | | 1500 | |
| Learning rate warmup ratio | | 1e-6 | |
| Optimizer | | AdamW (Loshchilov & Hutter, 2019) | |
| Adam beta1 | | 0.9 | |
| Adam beta2 | | 0.999 | |
| Adam weight decay | | 0.01 | |
| SHADE | False | True | True |
| RCS (Hoyer et al., 2022a;b) | | False | |

**Algorithm 1** PyTorch-like Pseudocode of Concept Sensitivity

```
# pipe: text-to-image generation diffusers pipeline
# c: str = "photorealistic first-person urban street view"
# c_augs: List[str] = List of the augmented prompts
# t: int = pre-defined timestep
# n_img: int = number of generated images for average

unet = pipe.unet
unet = unet.requres_grad_(True)
optimizer = torch.optim.AdamW(list(filter(lambda p: p.requires_grad,
unet.parameters())))) # optimizer for clear gradients

imgs = [pipe(c).images[0] for _ in n_img] # generate images

sensitivity = []

for img in imgs: # average over generated images
  for c_aug in c_augs: # average over augmented captions
      latent = pipe.vae.encode(img)
      noise = torch.randn_like(latent)
      noisy_latent = pipe.scheduler.add_noise(latent, noise, t)

      prompt_embeds = encode_prompt(c)
      model_pred = unet(noisy_latent, t, prompt_embeds)

      gt_diff = noise

      with torch.no_grad():
          prompt_embeds_aug = encode_prompt(c_aug)
          gt_concept = unet(noisy_latent, t, prompt_embeds_aug)

      loss_diff = torch.nn.functional.mse_loss(model_pred, gt_diff)
      loss_concept = torch.nn.functional.mse_loss(model_pred, gt_concept)

      loss_diff.backward(retain_graph=True)
      grads_diff = get_unet_grads(unet) # Algorithm 2
      optimizer.zero_grad()

      loss_concept.backward()
      grads_concept = get_unet_grads(unet) # Algorithm 2
      optimizer.zero_grad()

      sensitivity.append(grads_concept / grads_diff)
sensitivity_avg = average_gradients(sensitivity) # Algorithm 2
```

**Algorithm 2** PyTorch-like Helper Functions for Concept Sensitivity

```python
def getattr_recursive(module, attrs: List[str]):
    target_module = module
    for attr in attrs:
        target_module = getattr(target_module, attr)
    return target_module

def get_unet_grads(unet):
    grads = {'to_q': [], 'to_k': [], 'to_v': [], 'to_out.0': []}
    for attn_name in unet.attn_processors.keys():
        attn_module = getattr_recursive(unet, attn_name.split('.')[:-1])

        for proj_name in grads.keys():
            proj = getattr_recursive(attn_module, proj_name.split('.'))
            head_dim = 1 if proj_name == 'to_out.0' else 0
            grads_chunk = torch.chunk(proj.weight.grad.cpu(),
            attn_module.heads, dim=head_dim)
            grads[proj_name].append([(grad ** 2).mean().sqrt().item() for
            grad in grads_chunk])

    return grads

    def average_gradients(grads):
        grad_avg = {'to_q': [], 'to_k': [], 'to_v': [], 'to_out.0': []}

        for key in grad_avg:
            for grad in grads:
                grad_avg[key].append(grad[key])
            grad_avg[key] = torch.mean(torch.tensor(grad_avg[key]), dim=0)

        return grad_avg
```

**Algorithm 3** PyTorch-like Pseudocode of Modifying forward function of Selective LoRA

```python
# F: torch.nn.functional

def modify_to_selective_lora(layer, reduced_layer):
    # layer: diffusers.models.LoRACompatibleLinear
    # reduced_layer: 'A' or 'B'

    def selective_lora_set_lora_layer(self: LoRACompatibleLinear):
        def set_lora_layer(lora_layer, indices):
            self.lora_layer = lora_layer
            if indices is not None:
                self.indices = indices
        return set_lora_layer

    def selective_lora_set_forward(self: LoRACompatibleLinear):
        def forward(hidden_states: torch.Tensor, scale: float = 1.0):
            if self.lora_layer is None:
                return F.linear(hidden_states, self.weight, self.bias)
            else:
                if self.indices is not None:
                    # Selective LoRA (start)
                    org = F.linear(hidden_states, self.weight, self.bias)
                    if reduced_layer == 'B':
                        org[:, :, self.indices] += scale * \
                        self.lora_layer(hidden_states)
                    else:
                        org += scale * self.lora_layer(hidden_states[:, :, \
                        self.indices])
                    return org
                    # Selective LoRA (end)
                else:
                    return F.linear(hidden_states, self.weight, self.bias) + \
                    scale * self.lora_layer(hidden_states)
        return forward

    layer.set_lora_layer = selective_lora_set_lora_layer(layer)
    layer.forward = selective_lora_set_forward(layer)

    return layer
```

**Algorithm 4** PyTorch-like Pseudocode of selecting projection layers for Selective LoRA

```python
def apply_selective_lora(unet, selected_layers, rank):
    for attn_processor_name in unet.attn_processors.keys():
        _selected_layers = [selected_layer for selected_layer in
        selected_layers if '.'.join(attn_processor_name.split('.')[:-1])
        in selected_layer]
        if len(_selected_layers) == 0:
            continue
        attn_module = getattr_recursively(unet, attn_processor_name.split('.')[:-1])
        # getattr_recursively: Algorithm 2
        dim_head = attn_module.out_dim // attn_module.heads
        for layer_type in ('to_q', 'to_k', 'to_v', 'to_out.0'):
            selected_layers_proj = [selected_layer for selected_layer in
            _selected_layers if layer_type in selected_layer] is_out =
            layer_type == 'to_out.0'
            if len(selected_layers_proj) == 0:
                continue
            projection_layer = getattr_recursively(attn_module,
            layer_type.split('.'))
            # getattr_recursively: Algorithm 2

            # Head-wise Selective LoRA (start)
            head_indices = sorted([int(selected_layer.split('.')[-1][1:]) for
            selected_layer in selected_layers_proj])
            indices = sum([list(range(dim_head * head_idx, dim_head *
            (head_idx + 1))) for head_idx in head_indices], [])
            # Indices are split grouped by the dim_head
            # Head-wise Selective LoRA (end)

            projection_layer = modify_to_selective_lora_linear(projection_layer,
            reduced_layer='A' if is_out else 'B')
            # modify_to_selective_lora_linear: Algorithm 3

            if is_out:
                projection_layer.set_lora_layer(
                  LoRALinearLayer(
                    in_features=len(indices),
                    out_features=projection_layer.out_features,
                    rank=rank),
                  indices)
            else:
                projection_layer.set_lora_layer(
                  LoRALinearLayer(
                    in_features=projection_layer.in_features,
                    out_features=len(indices),
                    rank=rank),
                  indices)
    return unet
```

