# OpenReview forum: "Selective LoRA for Domain-Aligned Dataset Generation in Urban-Scene Segmentation"
_ICLR.cc/2025/Conference — Submitted to ICLR 2025_

### Official Review · Reviewer_kFPj · 2024-11-02

**Soundness:** 3
**Presentation:** 1
**Contribution:** 3
**Rating:** 6
**Confidence:** 5

**Summary:**

This paper proposes a novel approach to address the data scarcity in semantic segmentation by generating datasets (image-mask pairs) using text-to-image models. To solve the issue, it is necessary that the generative images align with the target doman and provide useful information beyond the training dataset. Therefore, this paper intorudces Selective LoRA, a finetuning approach for the pretrained text-to-image model that preserves the distributional diversity of the original pretrained model while aligning with the target domain. The proposed method selectively update weights for key concepts, such as style and viewpoint, which need to be aligned or to maintain the diversity. The authors shows that the proposed method generates datasets with desired distribution via ablation studies and improves in both in-domain and domain generalization settings.

**Strengths:**

The proposed method differentiate the weights to align with target domain while preserving valuable information of the pretarined model. From this, the proposed method selectively fine-tunes the model. This approach effectively addresses challenges when to adapt large pretrained models to different domains, enabling better domain specific performance without losing the benefits of the pretrained features.

The data scarcity problem addressed in this paper is a critical issue not only for semantic segmentation but also for a variety of vision tasks. Therefore, the proposed technique for generating image and ground truth pair datasets can be considered a core technology in the advancement of deep learning.

**Weaknesses:**

I believe that the proposed method tackles an important problem and offers a reasonable approach to addressing the challenges, which I highly commend. However, there are some weaknesses to consider.

First, the method relies on Stable Diffusion, trained on a large dataset. Although leveraging the distributional diversity learned by this pretrained model is the motivation behind the approach, it inherently sets an upper bound on the applicability of the proposed method based on the knowledge of the pretrained model. This is a fundamental limitation.

Defining the desired concepts, identifying the critical parts of the architecture where these concepts are expressed, and retraining the model are all highly manual processes that depend heavily on individual characteristics of target data . To define the desired concepts, the user of this method must analyze the distributions of the pretrained model and the target domain, identify the differing concepts, and guide the process with appropriate text prompts. Additionally, finding the associated weights and determining their importance requires experimental work, which lacks standardized criteria.

Generating images aligned with the desired distribution is crucial, but creating high-quality masks to accompany these images is equally important for semantic segmentation models. This aspect has not been sufficiently addressed. While the current method leverages intermediate features, there could be consideration of various other ways to generate masks from the images. Given that the proposed method utilizes a model trained on a large dataset, it might also be worth exploring the use of models like SAM (Segment Anything Model) for mask generation (of course, there are lots of candidates). I am not requring additional experiments using SAM. It woud be beneficial to analyze the quality of the masks for the generated images.

From a presentation, the paper is challenging to read. Figures 2 and 3 are difficult to understand, and it is not easy to infer the intended meaning from the related sections in the text. Additionally, the paper mentions L_Concept, but the figures use terms like L_style and L_viewpoint, which are not defined in the main text, causing confusion. The authors should clarify this and revise the figures accordingly. More detailed explanations about the process of creating text prompts are also necessary.

The experimental setup, including the ablation study, is not sufficiently explained. For instance, in experiments like those in Table 3, it is unclear how extensive the generated dataset is and how it is used.

Overall, the paper needs to be written in a way that is easier to understand.

**Questions:**

I believe that the proposed method demonstrates sufficient technical novelty and shows effectiveness through its quantitative experimental results. However, I think it would be beneficial for the authors to revise certain aspects to make the paper easier to understand. Improving the clarity of the text would enhance the overall presentation of the work.

---

> ### Author Response · Authors · 2024-11-22
>
> Thank you for your constructive feedback. We have carefully addressed your concerns below, and the proposed changes have been fully integrated into the revised manuscript. We welcome any additional input you may have.
>
> > [W4.1] I believe that the proposed method tackles an important problem and offers a reasonable approach to addressing the challenges, which I highly commend. However, there are some weaknesses to consider.
>
> Thank you for acknowledging the significance of our proposed method and its approach to addressing the challenges. We sincerely appreciate your commendation. Regarding the weaknesses mentioned, we have carefully addressed them in the revised manuscript and provided additional clarifications and experiments where necessary. We believe these revisions strengthen the paper and hope they adequately address your concerns. Please let us know if there are any further aspects that require clarification or improvement.
>
> > **[W4.2] Limitation of Reliance on Pretrained Stable Diffusion Models**
>
> We agree with the reviewer’s observation that a key limitation of our work is its reliance on the knowledge embedded in the pretrained Stable Diffusion model.
> While the additional information is indeed constrained by the prior knowledge of the pretrained text-to-image generation model, we have demonstrated significant improvements in urban-scene segmentation across both in-domain and domain generalization tasks.
> Furthermore, we believe that the proposed segmentation dataset generation framework has the potential to harness the extensive prior knowledge of large-scale text-to-image generation models for semantic segmentation by employing a selective adaptation methodology.
> This aspect has been addressed in the revised manuscript in **Section 5: Conclusion and Future Work**, **L530–534**.
>
>
> > **[W4.3] Challenges in Manual Concept Definition and Model Adaptation**
>
> As the reviewer pointed out, the process of defining desired concepts can indeed be quite manual.
> However, our main argument is that in cases where the desired concept is clear, such as urban-scene segmentation, our methodology provides an effective approach to exclusively learn the desired concept.
> Extending this approach to identify and adapt desired concepts for other target datasets, as we have demonstrated for urban-scene segmentation, could be an exciting topic for future research.
>
> Furthermore, the following newly conducted experiments demonstrate the robustness of our method:
>
> **[Pascal VOC]** For in-domain tasks, even beyond urban-scene datasets, training on Pascal-VOC using the identified style-sensitive layers effectively captured the style of Pascal-VOC, resulting in additional performance improvements. These results are detailed in **Figure 13** and **Table 9** in **Appendix A.8: In-Domain Experiments for the General Domain Dataset (Pascal-VOC)**.
>
> **[Robustness to Prompt Augmentation]** As shown in **Figure 12** in **Appendix A.7: Concept Sensitivity According to Prompt Augmentation**, we found that when the desired concept (style or viewpoint) is well-defined, the sensitivity score remains consistent across various prompt augmentations, highlighting the robustness of our approach.
>
> > Additionally, finding the associated weights and determining their importance requires experimental work, which lacks standardized criteria.
>
> As the reviewer noted, determining the proportion of selected layers may involve experimental work. However, we have introduced metrics for evaluating the quality of the generated dataset, including CMMD, CLIP-Score, and Image-Label Alignment (mIoU), as presented in **Table 3**, **Table 4**, and **Table 8**, which may serve as reasonable criteria.
> Additionally, we believe that the proportion of selected layers can be treated as a hyperparameter to adjust the exclusiveness of the desired concept relative to other concepts. We have also provided qualitative results and a detailed analysis in **Figure 16**, **Appendix A.10: Additional Qualitative Results**, and **L1241-L1273**, respectively.

---

> > ### Author Response · Authors · 2024-11-22
> >
> > > **[W4.4] Importance of Mask Quality and Alternative Mask Generation Methods**
> >
> > As the reviewer pointed out, creating high-quality masks corresponding to each generated image is indeed crucial.
> > Improving the label generator or exploring entirely different methodologies, such as SAM, is a promising future work direction.
> > While we utilize the architecture of the label generator in DatasetDM, we made a significant advancement in image-label alignment even with the same architecture.
> > We have added a new qualitative comparison in **Figure 6** of the revised manuscript, demonstrating that our Style-Selective LoRA significantly outperforms both DatasetDM and the original LoRA (which applies LoRA parameters to all layers).
> >
> > Furthermore, although we could not elaborate on this in the main paper due to the page limit, we initially included a discussion on image-label alignment in the Supplementary Material **Appendix A.6: Comparison of Image-Label Alignment**.
> > **Table 8** reports the significant improvements our method achieved over DatasetDM in terms of image-label alignment.
> > To provide quantitative results, we use the predictions from the pretrained Mask2Former model, which was fully supervised on the 100\% Cityscapes dataset and achieves a 79.40 mIoU, as a proxy for the ground truth mask.
> > Below are the key results from Table 8, which compares the image-label alignment (mIoU) of DatasetDM and ours under in-domain experiments. We selected 2\% of layers for Style-Selective LoRA.
> >
> > |  Method   | Image-Label Alignment |
> > | :-------: | :-------------------: |
> > | DatasetDM |         25.18         |
> > |   Ours    |         39.37         |
> >
> > Similar to the qualitative results, the Style-Selective LoRA significantly outperforms DatasetDM.
> > We attribute this to the domain gap between the pretrained text-to-image model (SDXL) and the source dataset (Cityscapes), as detailed in **Appendix A.6: Comparison of Image-Label Alignment** under **Analysis of the Qualitative Comparison**.
> >
> >
> > > **[W4.5] Improving Readability, Consistency in Figures, and Text Prompt Explanation**
> >
> > Acknowledging the importance of improving the presentation quality, we have made a non-trivial number of changes, including Figures 2 and 3 and other notations. Specifically,
> >
> > - In **Figure 2**, we focused on explaining our segmentation dataset generation framework using "viewpoint" as the target concept instead of explaining with both viewpoint and style.
> > - In **Figure 2**, the caption mentions each stage and its related section, which we reiterate in Section 3.1 when discussing the overall framework.
> > - In **Figure 3**, we 1) modified $L_{style}$ and $L_{viewpoint}$ to $L_{Concept}$ and 2) referenced the equation to help readers connect each symbol with the equation.
> > - We simplified the illustration in **Figure 3 (a)** by adding a note that "style-sensitive" layers react when the primary concept ("style") is generated differently.
> > - In **Figure 3 (b)**, we also added symbols for better understanding.
> >
> > Thank you for specifically pointing out the shortcomings in the presentation of our initial submission. If you have any further suggestions for the current revised version, we would be eager to incorporate them.
> >
> > > **[W4.6] Clarification of Experimental Setup and Ablation Study Details**
> >
> > We revised the experimental setup, including the ablation study, and organized it into **Section 4 Experiments**. Also, we added details of the datasets including their sizes and usages in **4.1. Experimental Setup** under **In-domain semantic segmentation**.
> >
> > The main updates are as follows:
> > - For the dataset sizes, we use 500 images for few-shot experiments (0.3\% -- 10\%) and 3,000 images for fully-supervised experiments.
> > - We train Mask2Former with few-shot samples, which we use as the baseline model.
> > - Then, we fine-tune this baseline model on various synthetically generated datasets using methods such as DatasetDM and ours.
> >
> > Furthermore, we modified **the Table 3 of the initial submission (now Table 1 in the revised version)**, and its caption to better describe the details of the in-domain segmentation performance:
> >
> > - Added caption: In the first row, we trained Mask2Former on various fractions of the Cityscapes dataset (Baseline). Then, we fine-tuned the baseline on DatasetDM and our generated datasets with 30K iterations and evaluated the performance of the fine-tuned segmentation models. Additionally, we include an additional fine-tuned baseline (Baseline (FT)) that is solely fine-tuned on the same real dataset for a fair comparison in terms of the total iterations.
> > - We modified the term "Real FT" to "Baseline (FT)" with specified types of training datasets to avoid confusion.
> > - We added a middle row stating, "For a fair comparison, we fine-tune the baseline for 30K iterations using the following datasets".
> > - We denoted the total iterations (e.g., 120K) to show that the baseline model was further fine-tuned for 30K iterations for each method.

---

> ### Author Response · Authors · 2024-11-22
>
> > **[Q4.1] Enhancing Clarity and Presentation for Better Understanding**
>
> We deeply appreciate the reviewer for recognizing the technical novelty and strengths of our experiments.
> Acknowledging the importance of improving the presentation of our paper, we have made the following revisions in **Sections 3 and 4**:
>
> - We revised **Figure 1** to highlight the issues found in previous studies. DatasetDM does not involve fine-tuning the pretrained T2I model on the source dataset, and we identified the overfitting problem when training all layers in the Original LoRA.
> - In the method section, we updated **Figure 3** to help understanding of **Section 3.2**.
> - The explanation of the segmentation dataset generation framework was insufficient. We added detailed explanations in **Section 3.1 Overall Framework** and **Section 3.4 Training Label Generator and Generating Diverse Segmentation Datasets**.
> - We reorganized the entire **Section 4. Experiments** section.

---

> > ### Comment · Reviewer_kFPj · 2024-11-24
> >
> > As you know, datasets like Cityscapes have distinct characteristics compared to general object detection datasets, presenting unique challenges such as class imbalance and co-occurrence issues[A]. These challenges make it one of the datasets with significant performance variations across different classes. I encourage analyzing these aspects in detail.
> >
> > A few key points to consider:
> >
> > 1. Class-specific Performance Analysis:
> > - Which classes show the most significant improvements?
> > - If the improved classes are rare or typically difficult to detect, this could strengthen the novelty of the method
> > - Please provide a table showing class-wise improvements and include a detailed analysis
> >
> > 2. Text Guidance Considerations:
> > Given that this is a text-guided approach, we might expect varying degrees of performance improvement based on different text variations. Have you considered using alternative class names beyond the provided prompts? How might this affect performance?
> >
> > 3. Domain Generalization Question:
> > I notice that the performance gains in domain generalization scenarios are lower compared to the improvements seen in Cityscapes. Could you explain the potential reasons for this discrepancy?
> >
> > The focus on these aspects, particularly the class-specific analysis and domain adaptation challenges, could provide valuable insights into the method's strengths and limitations.
> >
> > [A] Kim, D., Lee, S., Choe, J. and Shim, H., 2024, March. Weakly Supervised Semantic Segmentation for Driving Scenes. In Proceedings of the AAAI Conference on Artificial Intelligence (Vol. 38, No. 3, pp. 2741-2749).

---

> ### Author Response · Authors · 2024-11-25
>
> Thank you for your valuable suggestions. We present the following additional experiments, analyses, and discussions to address each of your considerations.
>
> > **1. Class-specific Performance Analysis**
>
> As the reviewer pointed out, measuring and analyzing class-wise IoU is a crucial aspect of urban-scene semantic segmentation. We have included the related experiments in **Appendix A.10. Class-wise Segmentation Performance Analysis**, with the key results summarized as follows.
>
> > Class-wise IoU for In-domain few-shot experiments (Cityscapes, 0.3\%)
>
> |     Class     | IoU (Improvements) | Pixel Proportions |
> | :-----------: | :----------------: | :---------------: |
> |      bus      |       +18.08       |      0.23\%       |
> |     fence     |       +10.99       |      0.87\%       |
> |    bicycle    |       +7.33        |      0.41\%       |
> |     wall      |       +5.21        |      0.66\%       |
> | traffic light |       +4.40        |      0.21\%       |
>
> We demonstrated particularly significant performance improvements for the aforementioned classes in **Figure 16**. As evident from the experimental results, as you mentioned, rare classes that appear in less than 1\% of the dataset showed substantial performance gains. This serves as an additional strength of our approach.
>
> In addition, we observed that certain classes were generated at proportions lower than their actual occurrence in the Cityscapes dataset.
> To address this, we proposed a methodology to generate additional samples for these underrepresented classes.
> As shown in **Figure 17**, unlike the Original LoRA, the proposed Selective LoRA methodology effectively generates additional samples for specific classes, as it exclusively learns the style of the source dataset.
>
> We conducted an experiment aimed at enhancing the performance of a specific target class (e.g., "person") by generating additional samples for that class.
> The result of the experiment is presented in **Figure 16**, which demonstrates that the additional dataset achieves more balanced performance across classes by adjusting the label proportions.
> The following highlights the key results of the performance improvements.
>
> | Method | Proportion of Person (\%) | IoU (Person) | mIoU |
> | ------ | :-----------------------: | :----------: | :--: |
> | Baseline | 1.22 | 64.77 | 41.83 |
> | Ours | 0.12 | 51.19 | 44.13 |
> | Ours + (Additional Person Dataset) | 0.82 | 61.43 | 44.59 |
>
> Please refer to **Segmentation Dataset Generation Focused on a Specific Class** in line 1283 for further details.
>
> > 2. **Text Guidance Considerations**: Given that this is a text-guided approach, we might expect varying degrees of performance improvement based on different text variations. Have you considered using alternative class names beyond the provided prompts? How might this affect performance?
>
> Applying text augmentations to the given class names to create more diverse text prompts for dataset generation is a highly promising research direction for our method.
> To explore this, we conducted experiments detailed in **Appendix A.11 Generating Datasets with Diverse Class Names**, where we replaced the simple use of "Car" for generating cars with more detailed class names. The key findings are summarized as follows.
>
> First, as shown in **Figure 18**, we augmented the text prompts by diversifying "Car" into more specific categories such as "SUV car", "Sedan car", "Convertible car", and "Hatchback car" for data generation. Our Selective LoRA demonstrated its ability to generate these diverse classes effectively, as it avoids overfitting to the Cityscapes content, maintaining its generalization capability.
>
> We then incorporated these generated samples into the dataset and conducted training with the augmented dataset, as shown in **Table 10**.
>
> | Method | IoU (Car) | IoU (Bus) | IoU (Motorcycle) | mIoU |
> | ------ | :-------: | :-------: | :--------------: | :--: |
> | Baseline | 84.02 | 12.51 | 16.11 | 41.83 |
> | Ours | 85.03 | 30.59 | 15.26 | 44.12 |
> | Ours + (Additional Diverse Car Dataset) | **85.34** | **32.48** | **17.77** | **44.95** |
>
> The generated dataset enhanced overall segmentation accuracy and demonstrated consistent performance improvements in vehicle classes such as car, bus, and motorcycle. Further exploration of additional test prompts presents a promising direction for future research.

---

> ### Author Response · Authors · 2024-11-25
>
> > 3. **Domain Generalization Question**: I notice that the performance gains in domain generalization scenarios are lower compared to the improvements seen in Cityscapes. Could you explain the potential reasons for this discrepancy?
>
> Thank you for recognizing the performance improvements in the in-domain setting. However, we believe the gains in the domain generalization (DG) setting are equally noteworthy.
>
> The effectiveness of our approach becomes even more evident when examining individual datasets rather than the average DG performance across four datasets.
> We observed significant improvements on datasets with adverse weather conditions.
> For example, our method enhanced the performance of ColorAug by 2.95 on ACDC and 4.06 on Dark Zurich, which represent substantial gains compared to existing baselines.
>
> Furthermore, on HRDA, one of the most advanced DG methods, the second-best baseline achieves a performance increase of 0.38 (with DGInStyle reporting only a slight improvement of 0.71), whereas our approach achieves a significantly higher improvement of 1.53.
>
> For the potential reasons why the performance improvements may appear modest, we would like to highlight the extensive use of color augmentations already employed by existing DG methods.
> As noted in **lines 454-460**, methods such as DAFormer and HRDA incorporate aggressive color augmentations. This extreme augmentation likely diminishes the impact of dataset generation approaches, including ours, by addressing color-related variations upfront.
>
> Nevertheless, while image-to-image translation-based methods such as DGInStyle and InstructPix2Pix show increasingly marginal gains on DAFormer and HRDA, our method, though slightly reduced, continues to deliver substantial improvements. We believe this underscores the robustness and effectiveness of our approach in addressing domain generalization challenges.

---

> ### Comment · Reviewer_kFPj · 2024-11-27
> **Re: "person" class and text guidance**
>
> I have an opinion about the text guidance and the performance degradation or minor improvement of the class "person".
> As the authors used, it would be helpful to consider specific and diverse words like "SUV" and "Sedan". However, I think it will be better for authors to consider more text guidacne for the class "person".
>
> The word "person" is fairly appropriate word to represent the class, because it represents human without any criteria. But it is too neutral for the distribution of the diffusion models' training data. Also, it is too ambiguous because of the class "rider" (in case of Cityscapes). Since "person" contains "rider", the class "person" needs to be changed into more specific words or clauses. It would be helpful to consider the word "pedestrian" which does not overlap with "rider".
>
> However, this is also another research direction. The authors don't have to consider this opinion. This will not affect my rating.

---

> > ### Comment · Reviewer_kFPj · 2024-11-27
> > **Re: Re: "person" class and text guidance**
> >
> > It is necessary to re-define class words in order to effectively guide the text-related models. Just using given class names is not the only answer. Although some reviewers might think there is no technical novelty, this approach will be the quickest approach to adapt for real-world or more practical scenario. (You can modify the word rider into "person riding a bicycle/motorbike"?).

---

> > > ### Author Response · Authors · 2024-11-28
> > >
> > > We sincerely appreciate your thorough review and valuable feedback on our paper. We hope our responses address your concerns effectively.
> > >
> > > The main contribution of our paper is a novel fine-tuning method using Selective LoRA, which selectively learns only the concepts needed for data generation while preventing overfitting. This approach enables the generation of domain-aligned and diverse images without requiring extensive text prompt engineering for data augmentation.
> > >
> > > We acknowledge your highly valuable opinion regarding prompt augmentation and recognize the importance of additional prompt augmentation for the person category. We believe that we have already provided a potential solution with the car example in **A.11. Generating Datasets with Diverse Class Names**. Additionally, we have chosen to leave this idea as future work because it is currently slightly outside the focus of this paper (selective fine-tuning method).
> > >
> > > We have addressed all the issues raised in your initial review, including extensive revisions and updates to figures and tables for better presentation. Additionally, we have responded to all concerns from the second review. We kindly ask the reviewer to take these efforts into consideration and hope that our revisions meet your expectations.
> > >
> > > Thank you for your time and consideration.

---

### Official Review · Reviewer_KMEN · 2024-11-03

**Soundness:** 3
**Presentation:** 3
**Contribution:** 3
**Rating:** 5
**Confidence:** 4

**Summary:**

This paper addresses the challenge of data scarcity in semantic segmentation by generating datasets through fine-tuned text-to-image generation models. Existing methods often overfit and memorize training data, limiting their ability to generate diverse and well-aligned samples. This paper proposes Selective LoRA that selectively identifies and updates only the weights associated with necessary concepts for domain alignment while leveraging the pretrained knowledge of the image generation model to produce more informative samples.
The authors demonstrate its effectiveness in generating datasets for urban-scene segmentation.

**Strengths:**

1. The paper is well-written and easy to follow.
2. The idea of using concept loss to find important weights is reasonable.
3. The ablation studies and analytical experiments are interesting and inspiring.

**Weaknesses:**

1. The proposed method outlines how to fine-tune LoRA for generating informative target-style images. However, the definition of "informative samples" is not clear. This lack of clarity may hinder the reader's understanding of the intended contributions. For example, Figure 1 would benefit from including examples of informative data to provide a clearer context for what constitutes an informative sample.

2. In Figure 1(b), the results of the LoRA-finetuned images for both foggy and night-time conditions appear remarkably similar, suggesting that the fine-tuning process may not have effectively generated between these two target styles. It raises concerns about the method's capability compared to the pretrained approach.

3. The proposed Selective LoRA generates images in a specific style and containing particular content, but it lacks a comparative analysis with existing text-driven diffusion models, such as Instruct-Pix2Pix. A comparison in terms of both the quality of generated images and adaptation performance would significantly enhance the paper's contributions and provide the reader with a clearer understanding of how the proposed method stands in relation to established techniques.

4. I find the results presented in Table 3 somewhat confusing. If I understand correctly, the baseline results are derived from fine-tuning Mask2Former using generated images, while RealFT represents the results from fine-tuning on real data. However, it is unclear how the authors obtained the labels for the generated data. Were these results obtained through an unsupervised training approach, or was an additional decoder trained similarly to the DatasetDM?

**Questions:**

Additional minor questions are listed as follows.
 1.  The method uses selective LoRA to solve the problem of data scarcity in cross-domain segmentation. However, there are some similar methods to select LoRA weights in other fields, like LoRA-SP [3], GS-LoRA [2], Tied-LoRA [1] etc.. The authors should discuss these papers.

[1] Tied-LoRA: Enhancing parameter efficiency of LoRA with Weight Tying

[2] Continual Forgetting for Pre-trained Vision Models

[3] LoRA-SP: Streamlined Partial Parameter Adaptation for Resource-Efficient Fine-Tuning of Large Language Models

2. The authors should compare more text-driven or image-driven generated dataset baselines, such as Instruct-Pix2Pix [4], PTDiffSeg [5], DATUM [6], etc.

[4] InstructPix2Pix: Learning to Follow Image Editing Instructions

[5] Prompting Diffusion Representations for Cross-Domain Semantic Segmentation

[6] One-shot Unsupervised Domain Adaptation with Personalized Diffusion Models

---

> ### Author Response · Authors · 2024-11-22
>
> Thank you for your valuable feedback. We have addressed your concerns in detail below, and these revisions have been reflected in the updated manuscript. If there are any additional points you'd like us to consider, please let us know.
>
> > **[W3.1. W3.2] Clarification of "Informative Samples" Definition and Figure 1**
>
> We acknowledge the lack of clarity regarding "informative samples". In response to the reviewer's recommendation, we revised Figure 1 and its caption, adding our informative examples to clearly illustrate the concept. Additionally, we updated **L78-L80** in the revised version to explicitly define 1) domain-aligned and 2) informative samples using the example provided in **Figure 1**.
> Below is a brief clarification of the informative sample.
>
> Informative samples refer to generated image-label pairs that provide additional information beyond what is available in the existing source dataset, as described in **L43–44**. In this study, the additional information is provided by pretrained text-to-image generation models (e.g., Stable Diffusion).
> The definition of "informative samples" varies depending on the problem setting in urban-scene segmentation. For example, when the target domain is ACDC, which includes driving-scene viewpoints and adverse weather, but the training dataset is Cityscapes, which consists of driving-scene viewpoints with only clear-day conditions, a diverse weather conditional dataset (informative) combined with a driving-viewpoint (domain-aligned) could serve as the optimal dataset for the problem.
>
> As illustrated in the revised **Figure 1**, learning and fixing the style of the source dataset (e.g., the clear-day style from Cityscapes) does not add additional information, making it uninformative for this scenario. However, it may still be considered informative in the context of in-domain segmentation. Therefore, we propose a flexible method that selectively learns only the viewpoint or style from the Cityscapes training data while avoiding overfitting to the other concepts.
>
>
> > **[W3.3] Comparative Analysis with Image-Driven Diffusion Models (InstructPix2Pix)**
>
> Comparing our method to the suggested baseline, InstructPix2Pix, is a critical experiment.
> We have conducted additional comparisons with InstructPix2Pix and included the results in **Table 2, Section 4.2, of our revised manuscript.**
>
> InstructPix2Pix is an image-to-image translation model that generates diverse styles with a given image, similar to DGInstyle, which was included as a baseline method in our initial manuscript.
> Consequently, the segmentation maps remain consistent across various generated images derived from a single image. Since InstructPix2Pix is designed to generate diverse styles of a given image, we concluded that it is not suitable for dataset augmentation in in-domain scenarios. To this end, we compared our proposed method with InstructPix2Pix using the domain generalization task using the domains of foggy, rainy, night-time, and snowy images in Cityscapes. The results are as shown.
>
> | DG Method | Generated Dataset |   ACDC    |    DZ     |    BDD    |    MV     |  Average  |
> | :-------: | :---------------: | :-------: | :-------: | :-------: | :-------: | :-------: |
> | ColorAug  |  InstructPix2Pix  |   56.02   |   26.92   |   54.03   |   60.44   |   49.35   |
> | ColorAug  |       Ours        | **56.07** | **29.75** | **54.35** | **61.40** | **50.39** |
> | DAFormer  |  InstructPix2Pix  |   55.13   |   26.93   |   54.61   |   62.36   |   49.76   |
> | DAFormer  |       Ours        | **55.83** | **31.68** | **54.68** | **63.09** | **51.32** |
> |   HRDA    |  InstructPix2Pix  |   58.50   |   29.56   |   56.10   |   64.10   |   52.07   |
> |   HRDA    |       Ours        | **58.93** | **34.41** | **56.56** | **64.54** | **53.61** |
>
> The table demonstrates that our proposed method consistently outperforms InstructPix2Pix across all DG methods. Specifically, we found that InstructPix2Pix is less effective for recent state-of-the-art baselines such as DAFormer or HRDA. This is because our method generates diverse scenes and their corresponding segmentation maps, whereas InstructPix2Pix merely changes the styles of a given image without creating diverse scenes. Such augmentation effects are already provided by severe data augmentation methods. This experiment clearly highlights the necessity of our proposed method, given the limitations of using current image-to-image translation models as data generators.

---

> > ### Author Response · Authors · 2024-11-22
> >
> > > **[W3.4] Clarification of Training Process and Baselines in Table 3**
> >
> > First, we found the details of stages 3 and 4 are neglected in our initial manuscript, including the segmentation label generation process.
> > We have newly included **Section 3.4: Training Label Generator and Generating Diverse Segmentation Datasets** to ensure our paper is self-contained, and revise minor elements of **Figure 2** for clearer presentation.
> > Additionally, we provide further implementation details for the label generator in **Appendix A.1**, accompanied by the newly added **Figure 7**, which illustrates the label decoder architecture. Below is a brief summary of the explanations for stages 3 and 4.
> >
> > **[Stage 3]** We train an additional lightweight label generator to produce a segmentation label corresponding to the image, following DatasetDM.
> > To train the label generator, we add noise to the given labeled image and denoise the image with the fine-tuned T2I model, which can provide semantically rich intermediate multi-level feature maps and cross-attention maps.
> > Distinct from DatasetDM, we train the label generator based on the fine-tuned T2I model using Selective LoRA.
> > The added fine-tuning process causes a significant difference in image-label alignment, which we discussed in **Appendix A.6 Comparison of Image-Label Alignment**.
> > Furthermore, due to the difference between the base T2I model, architecture details slightly changed, as described in **Appendix A.1**.
> >
> > **[Stage 4]** Diverse image-label pairs are generated to address both domain generalization and in-domain scenarios. For domain generalization, text prompts are modified to include adverse weather conditions (e.g., foggy, snowy, rainy, night-time) by extending the default prompt, such as "photorealistic first-person urban street view," to, for example, "in foggy weather," enhancing the model’s ability to generalize across varying environmental conditions. For in-domain scenarios, diversity is introduced by varying the class names within the prompt template (e.g., "… with car", "… with car", etc.), allowing for the generation of images that reflect different object class combinations while maintaining consistency with the in-domain characteristics.
> >
> > We modified **the Table 3 of the initial submission (now Table 1 in the revised version)**, and its caption to better describe the details of the in-domain segmentation performance:
> >
> > - Added caption: In the first row, we trained Mask2Former on various fractions of the Cityscapes dataset (Baseline). Then, we fine-tuned the baseline on DatasetDM and our generated datasets with 30K iterations and evaluated the performance of the fine-tuned segmentation models. Additionally, we include an additional fine-tuned baseline (Baseline (FT)) that is solely fine-tuned on the same real dataset for a fair comparison in terms of the total iterations.
> > - We modified the term "Real FT" to "Baseline (FT)" with specified types of training datasets to avoid confusion.
> > - We added a middle row stating, "For a fair comparison, we fine-tune the baseline for 30K iterations using the following datasets".
> > - We denoted the total iterations (e.g., 120K) to show that the baseline model was further fine-tuned for 30K iterations for each method.
> >
> > > **[Q3.1] Discussion of the Other Advanced LoRA Approaches**
> >
> > We sincerely appreciate the reviewer’s insightful suggestions regarding studies utilizing LoRA. We have clarified the relationship between our work and these studies **in lines 247 and 250 of the revised manuscript**. The details are as follows.
> >
> > While LoRA enables parameter-efficient fine-tuning of large-scale models, it does not provide a mechanism for specifying target learning concepts (\eg, style or viewpoint) from source datasets. Subsequent studies on LoRA have predominantly focused on enhancing the LoRA adapter itself (\eg, architectures), as seen in approaches like LoRA-SP, GS-LoRA, and Tied-LoRA. However, there has been limited exploration of identifying which layers are most effective for LoRA fine-tuning to learn specific target concepts, particularly in the context of urban-scene segmentation.
> > Thus, the advanced LoRA approaches recommended by the reviewer (LoRA-SP, GS-LoRA, and Tied-LoRA) take a complementary but distinct direction from our Selective LoRA. Nonetheless, we acknowledge that integrating these advanced LoRA methods with our Selective LoRA could be a beneficial direction for future research.

---

> > > ### Author Response · Authors · 2024-11-22
> > >
> > > > **[Q3.2] Comparative Analysis with Text- or Image-Driven Diffusion Models**
> > >
> > > As we responded in [W3.3], we compared our method with InstructPix2Pix, which we reported the results **in Table 2 of our revised manuscript.**
> > > Also, we included suggested studies [5] and [6] **in lines 158-160 of our revised paper.**
> > > However, we found that the two suggested studies focus on tasks distinct from the primary objective of our work.
> > >
> > > In addition to generating images, our main goal is to generate segmentation maps corresponding to a given image by fine-tuning a label generation module using a baseline model.
> > > In contrast, [5] is a model that directly performs segmentation, while [6] appears to focus on generating an unlabeled dataset and applying UDA without producing corresponding segmentation maps.
> > > This distinction makes them less suitable as direct baselines for our proposed method.
> > > Nevertheless, we acknowledge that both studies aim to enhance segmentation performance by leveraging the prior knowledge of pretrained text-to-image generation models, which aligns with the broader goals of our research.

---

> > > > ### Author Response · Authors · 2024-11-25
> > > >
> > > > We sincerely appreciate your thoughtful feedback and have revised the manuscript accordingly, addressing all the raised concerns. Additionally, we conducted the suggested experiments to further strengthen our contributions. We would greatly appreciate it if you could let us know whether these revisions and experiments adequately address your concerns and, if so, consider reflecting this in your evaluation.

---

> > > > ### Comment · Reviewer_KMEN · 2024-11-26
> > > >
> > > > Thank you for your efforts and the detailed response. The majority of my concerns have been addressed, but a few questions remain. Firstly, the generation of image-label pairs to enhance the perception model has already been tackled in DatasetDM. Therefore, I believe this contribution lacks sufficient novelty for ICLR publication. Regarding informative data generation, it plays a critical role in domain-alignment segmentation, and I recommend comparing your approach with those in [5] and [6], as both papers propose methods for diverse image generation.
> > > >
> > > > Additionally, based on my experience, improving segmentation quality in the target domain using generated images relies on two key factors: (1) domain-invariance of the generated images relative to the target domain, and (2) the quality of the corresponding pseudo labels. However, the paper does not provide a detailed analysis or discussion on these aspects.

---

> ### Author Response · Authors · 2024-11-28
>
> > Thank you for your efforts and the detailed response. The majority of my concerns have been addressed, but a few questions remain.
>
> Thank you for acknowledging our efforts and detailed responses. We are pleased to hear that the majority of your concerns have been addressed. Below, we provide responses to the remaining questions, including additional experiments and analyses.
>
> > **Clarification and Highlighting the Novelty of Our Contribution**
>
> Our contribution does not lie in the framework itself. As outlined in **Section 2.2. Segmentation Dataset Generation**, the use of generative models to create image-label pairs for enhancing perception models, has already been explored in prior works such as DatasetDM and DatasetGAN.
>
> What we aim to highlight, as summarized in **L099–L107**, is our proposal of the Selective LoRA fine-tuning method, which sets us apart from DatasetDM, as it directly uses pre-trained text-to-image generation models without any fine-tuning.
> Our method enables selective learning of desired concepts (e.g., style, viewpoint), as demonstrated in the updated **Figure 2, Stage 1 and 2**.
> We would like to emphasize that these two aspects represent reasonable and novel contributions unique to our work, as agreed upon by all reviewers (**B65c**, **hMNt**, **KMEN**, **kFPj**).
>
> > **Comparison with DATUM [6]**
>
> As per your suggestion, we conducted a comparison with DATUM [5] in the domain generalization setting.
> The following outlines the performance comparison between DATUM and our proposed method, which has been updated in **Table 2**.
>
> | DG Method | Generated Dataset |   ACDC    |    DZ     |    BDD    |    MV     |  Average  |
> | :-------: | :---------------: | :-------: | :-------: | :-------: | :-------: | :-------: |
> | DAFormer  |       DATUM       |   54.06   |   27.10   | **54.74** |   62.40   |   49.58   |
> | DAFormer  |       Ours        | **55.83** | **31.68** |   54.68   | **63.09** | **51.32** |
> |   HRDA    |       DATUM       |   58.11   |   30.18   | **56.94** |   64.29   |   52.38   |
> |   HRDA    |       Ours        | **58.93** | **34.41** |   56.56   |   **64.54**   | **53.61** |
>
> DATUM [5] proposes a One-shot UDA approach. One-shot UDA refers to a setting where a real image from each target domain ("foggy", "night-time", "rainy", and "snowy") is available for domain adaptation.
> In contrast, domain generalization assumes no access to any real images from the target domain.
> DATUM leverages a single real target domain image and uses a text-to-image generation model to create a diverse set of *unlabeled* images for the target domain.
> These *unlabeled* image sets are then used in combination with the given source domain by leveraging existing domain adaptation techniques (e.g., DAFormer, HRDA).
>
> On the other hand, as illustrated in **Figure 2. Stage 3**, our approach trains a label generator to create a *labeled* dataset, which is then directly mixed into the training set for further training.
> This fundamental difference allows our method to achieve significantly higher performance than DATUM, even though DATUM utilizes a real target domain image.
>
> Thank you for suggesting this comparison, as it provides an excellent opportunity to highlight the additional advantages of our approach.
>
> > **Comparison with PTDiffSeg [5]**
>
> We appreciate your comments and would like to address the clarification regarding the extra experiment involving PTDiffSeg.
>
> First, PTDiffSeg and our method have different research purposes. PTDiffSeg is a segmentation model designed to resolve domain generalization (DG) tasks, *not an image generation method*.
> In contrast, our goal is to develop a data generation method for the segmentation field that can produce domain-aligned and diverse image-label pairs.
> Specifically, PTDiffSeg introduces a novel architecture by utilizing a pretrained diffusion model as its backbone and compares it with other DG methods such as ColorAug, DAFormer, and HRDA.
> However, our method aims to resolve the lack of datasets for any existing segmentation model, whether in-domain or DG, to improve the segmentation performance.
> For example, our method can be applied to PTDiffSeg to achieve further improvements, similar to its application in DAFormer or HRDA.
> However, it can not offer a direct comparison between PTDiffSeg and our proposed approach.
>
> Second, unfortunately, PTDiffSeg's code is not currently publicly available, which causes challenges in conducting a quick comparison experiment. We are currently struggling with re-implementation. It is an intriguing experiment to investigate whether our generated dataset can effectively complement an advanced DG method like PTDiffSeg. However, we have concerns due to the re-implementation issue and the rebuttal schedule. Therefore, there is a possibility that we will include these results in our paper after the rebuttal period ends.

---

> ### Author Response · Authors · 2024-11-28
>
> > **Two Key Analysis for the Segmentation Dataset Generation**: (1) domain-invariance of the generated images relative to the target domain **(Image Domain Alignment)** and (2) the quality of the corresponding pseudo labels **(Image-Label Alignment)**
>
> We have already conducted both analyses in the revised manuscript for the in-domain setting.
> First, image domain alignment is discussed in the revised paper under **Section 4.3. Analysis**, specifically in the subsection titled **Image Domain Alignment**. As shown in **Table 3**, we provided quantitative results by measuring image domain alignment between the source dataset (Cityscapes) and the generated dataset using CMMD.
> Additionally, based on feedback from various reviewers, we analyzed image-label alignment qualitatively in **Figure 6** and quantitatively in **Table 8**. More detailed analyses can be found in **Appendix A.6 Comparison of Image-Label Alignment**.
>
> Thus, we interpreted the current reviewer’s question as requesting additional analyses in the domain generalization setting.
> To address this, we have included a new analysis in **Appendix A.12. Additional Analysis of Our Generated Dataset on the Domain Generalization Setting**, with the key results summarized as follows.
>
> > **Image Domain Alignment in Domain Generalization Setting**: (1) domain-invariance of the generated images relative to the target domain
>
> For domain generalization, we generated additional datasets for "foggy", "night-time", "rainy", and "snowy" conditions and incorporated them into the training process.
> To evaluate text adherence for each condition, we analyzed the datasets using CLIP scores, as shown in **Table 4**.
>
> In this additional analysis, we further investigate how well each generated image set aligns with the actual images from ACDC corresponding to "foggy", "night-time", "rainy", and "snowy" conditions.
> This evaluation is performed using CMMD to measure image domain alignment (newly added in **Table 11**).
>
> |     Method      | foggy | night-time | rainy | snowy | average |
> | :-------------: | :---: | :--------: | :---: | :---: | :-----: |
> |      DATUM†      | 2.41  |    2.46    | 2.91  | 2.10  |  2.47   |
> | InstructPix2Pix | 3.43  |    3.13    | 2.99  | 3.32  |  3.22   |
> |    DatasetDM    | 4.90  |    5.52    | 5.34  | 4.96  |  5.18   |
> |      Ours       | 2.43  |    2.55    | 2.62  | 2.63  |  2.56   |
>
> † *For DATUM, we provide an additional image for each weather condition to satisfy the requirements of the One-shot UDA setting, whereas the other methods do not rely on target domain images.*
>
> As demonstrated in the results, our generated dataset achieves significantly better image domain alignment for each adverse weather condition compared to DatasetDM and InstructPix2Pix.
> This improvement likely arises from our method, which exclusively learns viewpoint information from Cityscapes, effectively utilizing driving scene knowledge.
> By contrast, DATUM requires a single real ACDC image for each weather condition and trains *four separate models*, one for each condition, *using the additional target domain images*.
> Even without access to any real ACDC images and using a single model to generate datasets for multiple weather conditions, our approach achieves a comparable level of image domain alignment to that of DATUM.
>
> Additionally, while our method currently addresses in-domain and domain generalization settings, exploring its application in a One-shot UDA setting, where additional training is performed using a single provided target domain image, would be an intriguing direction for future work.

---

> ### Author Response · Authors · 2024-11-28
>
> > **Image-Label Alignment in Domain Generalization Setting**: (2) the quality of the corresponding pseudo labels.
>
> In this analysis, we aim to evaluate how reliably our method provides pseudo labels compared to other methods in the domain generalization setting, both qualitatively and quantitatively.
> The detailed analysis is provided in **Appendix A.12**, featuring **Figure 19**, **Table 12**, and **Table 13**.
>
> First, we added qualitative results in **Figure 19**, which show that our method provides better labels than DatasetDM or InstructPix2Pix.
> For quantitative results, we utilized pretrained segmentors, as done in the analysis presented in **Table 8**.
> However, the previously used pretrained Mask2Former, trained only on Cityscapes, does not deliver reliable performance under adverse weather conditions.
>
> To address this, we fine-tuned the pretrained Mask2Former (M2F) on the ACDC training set for each weather condition, creating specialized segmentors for "foggy", "night-time", "rainy", and "snowy".
> The performance of these segmentors on the ACDC validation set before and after fine-tuning is presented below.
>
> |   Segmentor    | foggy | night-time | rainy | snowy | average |
> | :------------: | :---: | :--------: | :---: | :---: | :-----: |
> | Pretrained M2F | 67.66 |   23.17    | 51.94 | 47.55 |  47.58  |
> | Fine-tuned M2F | 78.54 |   52.16    | 66.23 | 74.79 |  67.93  |
>
> Each fine-tuned model is now expected to provide more meaningful segmentation maps for its respective weather condition.
> The results of measuring Image-Label Alignment using these models are as follows.
>
> |     Method      | foggy | night-time | rainy | snowy | average |
> | :-------------: | :---: | :--------: | :---: | :---: | :-----: |
> | InstructPix2Pix | 25.98 |   48.60    | 63.04 | 40.66 |  44.57  |
> |    DatasetDM    | 40.84 |   35.90    | 47.43 | 44.02 |  42.05  |
> |      Ours       | 41.55 |   43.07    | 48.69 | 39.47 |  43.20  |
>
> As shown in the table, InstructPix2Pix, which directly uses Cityscapes labels and only slightly edits the weather conditions of the images, demonstrates an advantage in image-label alignment.
> Despite its high image-label alignment, we highlighted the limited performance improvements of InstructPix2Pix in **Table 2** and **Section 4.2 Main Results**, attributing this to the lack of scene diversity caused by its reliance on fixed segmentation label maps.
>
> When comparing methods that generate labels, our approach achieves better image-label alignment than DatasetDM.
> This improvement stems from our text-to-image generation model learning viewpoints from Cityscapes.
> As a result, even with the same label generator architecture, our finetuned text-to-image generation model provides representations with a smaller domain gap when generating images based on the Cityscapes dataset.
>
> We sincerely appreciate the opportunity to perform more diverse comparisons and analyses that highlight the strengths of our approach, as well as the chance to further clarify our contributions.
> If you have any additional questions or concerns, please do not hesitate to let us know.

---

> > ### Author Response · Authors · 2024-12-01
> >
> > As the discussion period approaches its conclusion, we would like to respectfully remind you of our revised manuscript. We kindly request your feedback on the second set of comments and would greatly appreciate your reconsideration of the evaluation in light of our updates.
> >
> > Thank you sincerely for your time and efforts.

---

### Official Review · Reviewer_hMNt · 2024-11-03

**Soundness:** 2
**Presentation:** 2
**Contribution:** 2
**Rating:** 5
**Confidence:** 5

**Summary:**

The paper presents a method for fine-tuning pre-trained T2I models to generate datasets specifically for urban-scene segmentation, addressing the challenge of data scarcity. Traditional methods often utilize pre-trained T2I models directly or apply LoRA for fine-tuning, which can lead to generated samples that fail to align with the target domain or lack diversity. To overcome these issues, the paper introduces Selective LoRA, a novel fine-tuning approach that selectively identifies and updates the weights that are most closely associated with specific concepts for domain alignment. This approach reduces the number of parameters that need training, improving training efficiency while ensuring that the original T2I model's generalizability is preserved. Extensive experiments demonstrate that the generated datasets improve the performance of previous segmentation models in urban-scene segmentation.

**Strengths:**

S1: The paper "Selective LoRA" introduces a new training strategy for fine-tuning pretrained T2I models to generate diverse datasets for segmentation tasks, addressing the challenge of data scarcity.
S2: Extensive experiments show that the generated datasets enhance the performance of prior segmentation models in urban-scene segmentation.

**Weaknesses:**

W1: The writing in this paper is somewhat challenging to understand, and the technical descriptions lack clarity, which can lead to confusion during reading. Below are a few examples, though not exhaustive. For instance, in Figures 3 and 4, what do the layer indices represent? Are they the projection layers for all attention in the network? However, according to the description in line 251, it seems that all linear layers in the network are being trained with LoRA. Additionally, Section 3 only covers the first two stages, with the third and fourth stages not being described in detail, making this part less clear. The structure of the experimental section is also somewhat disorganized.

W2: The design of the tables lacks standardization, leading to confusion for the reader. Here are a few examples, though not exhaustive. For instance, many tables do not clearly explain what the numerical values represent, making interpretation difficult. In Table 3, the baseline names should be listed directly. Additionally, the entries under the "Data Ratio" column correspond to various methods, which creates some confusion. Furthermore, for the methods used to generate datasets that enhance baseline performance in Table 3, it would be clearer to label them as "Baseline + Real FT" rather than just "Real FT."

W3: Additionally, I noticed that the baseline appears to be from a 2022 paper. Are there any more recent baselines available for comparison?
W4: Some modules may not appear particularly novel from a technical perspective. LoRA are also commonly used in various papers.

**Questions:**

Q1: From the design motivation, it seems that training all LoRA parameters may lead to overfitting, resulting in reduced diversity in the generated images. In contrast, Selective LoRA selects a subset of parameters that are most associated with the concepts, effectively training fewer parameters and better preserving the original T2I model's capabilities. The original LoRA setting applies training to all linear layers in the UNet with LoRA. I wonder if training LoRA only on certain layers' cross-attention (few parameters) could achieve a similar effect as Selective LoRA.

Q2: I hope the authors can address the concerns raised in the "Weaknesses" section.

---

> ### Author Response · Authors · 2024-11-22
>
> We greatly appreciate your thoughtful comments. A detailed response to your concerns is provided below, and the corresponding updates have been included in the revised manuscript. Please feel free to share any further suggestions.
>
> > **[W2.1] Enhancing Writing, Technical Descriptions, and Experimental Structure**
>
> We acknowledge the importance of improving the presentation of our paper and appreciate you pointing this issue out.
> We reorganized the paper and made overall adjustments to parts including **Sections 3 and 4, and Figures 1, 2, 3, 4 and 5.**
> Following are the clarifications regarding the issues that the reviewer is confused with.
>
> **[LoRA Fine-Tuning]** We conducted LoRA fine-tuning on Stable Diffusion XL by adding LoRA layers to *all attention linear projection layers (query, key, value, and output)*.
> Since "Original LoRA" in this work refers to updating all the added LoRA layers, we have revised the description in L249 in the revised manuscript to reflect this accurately.
> Additionally, we labeled all figures involving the T2I model with "Attention Layers" to clearly indicate that these are the attention layers of the T2I model, including Figures 2, 3, 4, 5, and others.
>
> **[Layer Indices]** The layer indices in Figures 3 and 4 represent the "attention layer indices" identified in Stable Diffusion XL. The left side corresponds to shallower layers, while the right side corresponds to deeper layers in a continuous progression. We labeled all relevant figures with "shallower" to "deeper" to provide clearer context.
>
> **[Stage 3, 4]** For stages 3 and 4, we have newly included **Section 3.4: Training Label Generator and Generating Diverse Segmentation Datasets** to ensure our paper is self-contained, and revise minor elements of **Figure 2** for clearer presentation.
> Additionally, we provide further implementation details for the label generator in **Appendix A.1**, accompanied by the newly added **Figure 7**, which illustrates the label decoder architecture. Below is a brief summary of the explanations for stages 3 and 4.
>
> **[Brief Summary of Stage 3]** We train an additional lightweight label generator to produce a segmentation label corresponding to the image, following DatasetDM.
> To train the label generator, we add noise to the given labeled image and denoise the image with the fine-tuned T2I model, which can provide semantically rich intermediate multi-level feature maps and cross-attention maps.
> Distinct from DatasetDM, we train the label generator based on the fine-tuned T2I model using Selective LoRA.
> The added fine-tuning process causes a significant difference in image-label alignment, which we discussed in **Appendix A.6 Comparison of Image-Label Alignment**.
> Furthermore, due to the difference between the base T2I model, architecture details slightly changed, as described in **Appendix A.1**.
>
> **[Brief Summary of Stage 4]** Diverse image-label pairs are generated to address both domain generalization and in-domain scenarios. For domain generalization, text prompts are modified to include adverse weather conditions (e.g., foggy, snowy, rainy, night-time) by extending the default prompt, such as "photorealistic first-person urban street view," to, for example, "in foggy weather," enhancing the model’s ability to generalize across varying environmental conditions. For in-domain scenarios, diversity is introduced by varying the class names within the prompt template (e.g., "… with car", "… with car", etc.), allowing for the generation of images that reflect different object class combinations while maintaining consistency with the in-domain characteristics.
>
> **[Organizing Experimental Section]**
> We have reorganized the experimental section (Section 4) as the reviewer suggested.
> To be more specific, we explained the experimental setup and implementation details at the beginning of the experimental section.
> Then, we reported the main performance improvements in a section titled 'Main Results on the Semantic Segmentation Benchmarks'.
> After the main result, we demonstrated various analyses and ablation studies.
> We deeply appreciate the reviewer's valuable feedback.

---

> > ### Author Response · Authors · 2024-11-22
> >
> > > **[W2.2] Standardization and Clarity in Table Design**
> >
> >
> > We modified **the Table 3 of the initial submission (now Table 1 in the revised version)**, and its caption to better describe the details of the in-domain segmentation performance:
> >
> > - Added caption: In the first row, we trained Mask2Former on various fractions of the Cityscapes dataset (Baseline). Then, we fine-tuned the baseline on DatasetDM and our generated datasets with 30K iterations and evaluated the performance of the fine-tuned segmentation models. Additionally, we include an additional fine-tuned baseline (Baseline (FT)) that is solely fine-tuned on the same real dataset for a fair comparison in terms of the total iterations.
> > - We labeled the first cell as "Method"
> > - We used Mask2Former as the baseline segmentation model.
> > - We modified the term "Real FT" to "Baseline (FT)" with specified types of training datasets to avoid confusion.
> > - We added a middle row stating, "For a fair comparison, we fine-tune the baseline for 30K iterations using the following datasets".
> > - We denoted the total iterations (e.g., 120K) to show that the baseline model was further fine-tuned for 30K iterations for each method.
> >
> > > **[W2.3] Clarification of Recent Baselines**
> >
> > We want to clarify that the baseline methods that we compare with are DatasetDM (NeurIPS 2023) and DGInStyle (ECCV 2024), not Mask2Former, ColorAug, DAFormer, or HRDA.
> > Additionally, at the request of reviewer KEMN during the rebuttal period, we compared our method with InstructPix2Pix (CVPR 2023) in **Table 2**, which generates images of diverse styles (e.g., adverse weather) with a fixed label map for a given image, under the domain generalization setting.
> > We believe that papers accepted to CVPR 2023, NeurIPS 2023, ECCV 2024 represent recent baseline methods for comparison.
> > To the best of our knowledge, we are not aware of other studies generating datasets for segmentation tasks beyond the baselines methods we have compared with.
> > If we are missing any recent seminal work, please inform us, and we will compare and discuss it in our revised paper.
> >
> > > **[W2.4] Clarification of Our Technical Novelty Distinct from LoRA**
> >
> > **In lines 254 to 256 of our revised paper**, we clarified the uniqueness and technical novelty of our proposed method compared to LoRA and its variants.
> > As the reviewer mentioned, LoRA is indeed widely used in various papers recently.
> > We want to clarify that the technical novelty of our paper does not come from using LoRA itself.
> > Instead, our novelty lies in identifying the specific parameters responsible for learning a desired concept and updating only those parameters for generating pairs of images and segmentation maps.
> > In other words, while LoRA has been widely adopted in various studies, the question of *how* to utilize LoRA for dataset generation in urban-scene segmentation has been underexplored.
> > We addressed this question by introducing the concept of Selective LoRA, providing valuable insights to the community.
> >
> > > **[Q2.1] Comparison of Selective LoRA and Hand-Crafted Layer Selection Approaches**
> >
> > We concur with the reviewer's idea that training LoRA only on certain cross-attention layers in a hand-crafted manner is a viable approach.
> > To demonstrate the superiority of our proposed method over such an approach, we conducted a comparison detailed in **Appendix A.9: Comparison with Hand-Crafted Layer Selection Approaches** in our revised paper.
> > The quantitative results are presented in **Figure 14**, and the qualitative results are shown in **Figure 15**. Below are the key findings from our quantitative results presented in **Figure 14**.
> >
> > Under the few-shot setting of Cityscapes (0.3\%), we compared our proposed method with the reviewer's suggestion applied to self-attention layers only (SA-only) and cross-attention layers only (CA-only).
> >
> > |  Fine-tuning target   | Proportion of the fine-tuned layers | Segmentation Performance (mIoU) |
> > | :--: | :--: | :--: |
> > | Pretrained | 0\% | 42.82 |
> > | SA-only | 50\% | 43.40 |
> > | CA-only | 50\% | 43.46 |
> > | Original LoRA | 100\% | 42.97 |
> > | **Ours** | 2\% | **44.13** |
> >
> > As shown in the table, our method achieves the best segmentation performance (mIoU) while updating a much smaller number of layers compared to other manually selected approaches or the original LoRA.
> > The main reason is that our method can identify the certain layers responsible for learning the desired concept, which is challenging to achieve by manually selecting the layers.

---

> > > ### Author Response · Authors · 2024-11-25
> > >
> > > Thank you once again for your valuable insights, which significantly helped us improve our work. We have carefully revised the paper and included the additional experiments that were requested. Could you kindly let us know if the updated manuscript resolves your concerns and, if possible, whether this might warrant a reconsideration of the score?

---

> ### Author Response · Authors · 2024-11-28
>
> As the discussion period is nearing its end, we wanted to kindly remind you about our revised manuscript.
>
> We have carefully revised the paper and incorporated the additional experiments you requested. Furthermore, we have addressed the major concerns raised by the other reviewer.
>
> Could you please let us know if our revisions resolve your concerns? Your feedback is crucial, and we sincerely hope our updates meet your expectations.
>
> Thank you for your time and consideration.

---

> > ### Comment · Reviewer_hMNt · 2024-12-02
> >
> > Thank you for the effort in significantly updating the paper. I am considering increasing my score; however, as shown in the new table provided by the authors, the improvement from selective LoRA is not as significant compared to fine-tuning only some of the LoRA layers.
> > Additionally, training a LoRA for a specific domain is a common practice and not particularly novel. Therefore, I can only increase my score to 5.

---

> ### Author Response · Authors · 2024-12-02
>
> Thank you for sharing your valuable feedback. Below, we provide additional responses to address the concerns raised by the reviewer.
>
> > however, as shown in the new table provided by the authors, the improvement from selective LoRA is not as significant compared to fine-tuning only some of the LoRA layers.
>
> We have further highlighted the degree of performance improvement achieved through LoRA fine-tuning based on the performance of DatasetDM, the major baseline that utilizes the pretrained text-to-image generation model.
> The table allows a direct comparison of the performance gains obtained purely through LoRA fine-tuning.
>
> |  Fine-tuning target   | Proportion of the fine-tuned layers | Segmentation Performance (mIoU) | Performance Improvement via LoRA fine-tuning (mIoU) |
> | :--: | :--: | :--: | :--: |
> | Pretrained (DatasetDM) | 0\% | 42.82 | - |
> | Original LoRA | 100\% | 42.97 | + 0.15 |
> | SA-only | 50\% | 43.40 | + 0.58 |
> | CA-only | 50\% | 43.46 | + 0.64 |
> | **Ours** | 2\% | **44.13** | **+ 1.31** |
>
> > **Table R2.1.** Comparison of Performance Improvements Across Various LoRA Fine-tuning Approaches
>
> As shown in the table, our Selective LoRA approach achieves a performance improvement of 1.31, which is *more than twice* the improvement observed with hand-crafted layer selection approaches (SA-only, CA-only).
> Furthermore, we believe that a 1.31 mIoU improvement is significant in the context of segmentation performance.
> Detailed experimental results can be found in **Appendix A.9: Comparison with Hand-Crafted Layer Selection Approaches** in our revised paper.
>
> > Additionally, training a LoRA for a specific domain is a common practice and not particularly novel.
>
> While LoRA fine-tuning for specific domains is widely recognized, as noted by the reviewer, no prior studies have explored its application in segmentation dataset generation approaches [1, 2, 3].
> Furthermore, as demonstrated in the comparison of LoRA fine-tuning methods provided in the response above (**Table R2.1**), we have shown that the Original LoRA only marginally improves performance due to issues such as overfitting and memorization, which results in the generation of non-informative samples.
> In contrast, our Selective LoRA achieves a significant performance improvement of 1.31. This clearly highlights the substantial difference between Selective LoRA and Original LoRA.
>
> The additional detailed technical differences with the original LoRA, recognized as a novel approach by all reviewers (**B65c**, **hMNt**, **KMEN**, **kFPj**), can be found in **L254-L256** of our revised paper, as outlined below.
>
> > **L254-L256**: The key distinction of Selective LoRA lies in *selectively fine-tuning only the crucial layers* based on an automatically computed score, termed concept sensitivity, for the desired concept in the source dataset, while previous LoRA studies update all projection layers.
>
> Thank you once again for providing valuable feedback. Although there isn’t much time remaining, if you have any additional concerns or questions, please let us know, and we will do our best to address them within the discussion period.
>
> Reference
>
> > [1] Wu, Weijia, et al. "Datasetdm: Synthesizing data with perception annotations using diffusion models." Advances in Neural Information Processing Systems 36 (2023): 54683-54695.
> > [2] Wu, Weijia, et al. "Diffumask: Synthesizing images with pixel-level annotations for semantic segmentation using diffusion models." Proceedings of the IEEE/CVF International Conference on Computer Vision. 2023.
> > [3] Nguyen, Quang, et al. "Dataset diffusion: Diffusion-based synthetic data generation for pixel-level semantic segmentation." Advances in Neural Information Processing Systems 36 (2024).

---

### Official Review · Reviewer_B65c · 2024-11-03

**Soundness:** 3
**Presentation:** 3
**Contribution:** 2
**Rating:** 6
**Confidence:** 4

**Summary:**

The paper introduces a new approach to generate training samples with sepecific concepts variants. The proposed method learn specific concepts, such as style or viewpoint, by selectively manipulate the gradients. The method claim that it improve domain alignment and sample diversity.  In experiments, the method are compared with baseline and the DatasetDM, and results show improvements in in-domain, few-shot segmentation. The wide-scope experiments prove its practicability.

**Strengths:**

1. The paper is well-organized, the motivation description is clear and concise. The pipeline figures are easy to understand.
2. The core idea is interesting, use language distinction to learn concept difference, and eliminate the requirement of paired visual data to learn specific concepts. I think the learning procedure is practicable.
3. The experimental settings are extensive, including both in-domain, few-shot and damain generalization.

**Weaknesses:**

1. The method mainly focus on introduce their Selective LoRA. However, the whole pipline includes training a label generator (stage 3 in Figure 2). The technical details in this part are not well delivered. How the label generator receive the intermediate features from T2I models and generate semantic maps?  In addition, in line 190-197, the authors say their use Mask2Former as label generator as same as DatasetDM. As I know, the DatasetDM use only "perceptual decoder" which only includes a decoder architecture instead of whole Mask2Former segmentaiton. Clarifying this distinction could provide a clearer understanding of the contributions of the current approach.
2. While the method aims for simultaneous sample and segmentation map generation, it requires a two-stage training process for the T2I model and label generator separately, contrasting with DatasetDM’s one-stage training. This additional stage could indeed limit practicality for real-time or large-scale augmentation, and a comparison in training efficiency or practical adaptability would be beneficial.
3. The dataset used for evaluation is Cityscapes and BDD100K, which includes only city streets. Since singlar scene makes learning specific concepts changes easiler, it would be improved if the authors prove their method on more general dataset, e.g. coco, ade20k. Since the main comparison method DatasetDM use more general dataset, I wonder performances of Selective LoRA on other datasets.
4. If the selective learning process affects the reliability of generated segmentation maps? The authors seem not provide relavant discussion.
5. Minor erros: the box of viewpoints and styles in stage 4) of Figure 2 are reversed.

Reference:
[1] DatasetDM: Synthesizing Data with Perception Annotations Using Diffusion Models, NIPS 2023

**Questions:**

Please see Weaknesses. If the authors address my concerns, I will raise my rate.
Additionally, It would be better if:
1. Provide quantitative metrics on the quality of the generated segmentation maps, comparing them to ground truth or to maps generated by other methods. Discuss any observed differences in segmentation map quality between their method and baseline approaches, particularly in relation to the selective learning process.
2. If possible, include a qualitative analysis (e.g., visual examples) of any artifacts or inconsistencies in the generated segmentation maps that might be attributed to the selective learning process.

---

> ### Author Response · Authors · 2024-11-22
>
> We sincerely appreciate your insightful feedback. Detailed responses to your concerns are provided below, and the revisions have been incorporated into the updated version. Please let us know if there are any further suggestions or comments.
>
> > **[W1.1] Clarification of Label Generator Integration and Method Distinction**
>
> The explanation of the label generator was provided in **Appendix A.1** of the original paper. However, as the reviewer pointed out, the main paper did not sufficiently explain Stage 3 (label generator). Additionally, the description of Stage 4 (generating image-label pairs) was inadequate. Therefore, we added "**Section 3.4: Training the Label Generator and Generating Diverse Segmentation Datasets** to explain both Stage 3 and Stage 4. Furthermore, we extended **Appendix A.1** with additional technical details and added **Figure 7** to illustrate the label decoder architecture.
> Below is a brief summary of the explanations for stages 3 and 4.
>
>
> **[Stage 3]** We train an additional lightweight label generator to produce a segmentation label corresponding to the image, following DatasetDM.
> To train the label generator, we add noise to the given labeled image and denoise the image with the fine-tuned T2I model, which can provide semantically rich intermediate multi-level feature maps and cross-attention maps.
> Distinct from DatasetDM, we train the label generator based on the fine-tuned T2I model using Selective LoRA.
> The added fine-tuning process causes a significant difference in image-label alignment, which we discussed in **Appendix A.6 Comparison of Image-Label Alignment**.
> Furthermore, due to the difference between the base T2I model, architecture details slightly changed.
>
> Specifically, given the increased number of blocks and channels in SDXL, we selected specific blocks to extract multi-scale feature maps and multi-scale cross-attention maps.
> Feature maps were extracted from the last feature block at each resolution of the upsampling blocks, while cross-attention maps were sampled at regular intervals (every 7 blocks) from the total 36 upsampling blocks (i.e., 1st, 8th, ... 29th, 36th).
> Moreover, as shown in the updated Figure 7, Stable Diffusion XL has only three resolution levels, compared to the four resolution levels in the Stable Diffusion v1.5 architecture. This difference required minor adjustments to the pixel decoder and transformer decoder, as described in **Appendix A.1**.
> *Importantly, to ensure a fair comparison, the reported scores for DatasetDM were obtained using a re-implemented version based on SDXL with the same modifications.*
>
>
> **[Stage 4]** Diverse image-label pairs are generated to address both domain generalization and in-domain scenarios. For domain generalization, text prompts are modified to include adverse weather conditions (e.g., foggy, snowy, rainy, night-time) by extending the default prompt, such as "photorealistic first-person urban street view," to, for example, "in foggy weather," enhancing the model’s ability to generalize across varying environmental conditions. For in-domain scenarios, diversity is introduced by varying the class names within the prompt template (e.g., "… with car", "… with car", etc.), allowing for the generation of images that reflect different object class combinations while maintaining consistency with the in-domain characteristics.
>
> > **[W1.2] Comparison in Training Efficiency with DatasetDM**
>
> As mentioned in lines L402-404 of the revised manuscript, training Selective LoRA only requires *one hour* on a single Tesla V100 GPU.
> While the reviewer is concerned about the additional stage in our method, we want to clarify that the additional stage requires a marginal amount of time compared to the entire training time (20 hours for training label generator).
> Considering the performance gains of using Selective LoRA compared to DatasetDM, we believe that the additional one hour of training is not as significant as the reviewer may be concerned about.

---

> ### Author Response · Authors · 2024-11-22
>
> > **[W1.3] Semantic Segmentation Experiments on More General Datasets**
>
> In response to the reviewer's request, we conducted a new experiment for generating a semantic segmentation dataset using Pascal-VOC, a more general dataset in **A.8 In-domain Experiments for the General Domain Dataset (Pascal-VOC)**.
> For this, we utilized only 100 images for segmentation, followed by a few-shot Pascal-VOC experiment in DatasetDM.
>
> > Pascal-VOC (100 images)
>
> | Generated Dataset                        | mIoU  | Improvement |
> | ---------------------------------------- | :---: | :---------: |
> | Baseline (Mask2Former)                   | 44.59 |      -      |
> | **Fine-tuning with additional datasets** |       |             |
> | DatasetDM                                | 36.16 |     - 8.43     |
> | Ours         | 45.52 |    + 0.93     |
>
> As shown in the table, although DatasetDM reduces performance compared to the baseline, our approach successfully enhances segmentation performance, even on a more general dataset. This experiment demonstrates the effectiveness of the selective learning approach, even with datasets exhibiting minimal distribution shifts.
> We include further experimental details and a qualitative comparison in **Appendix A.8** and **Figure 13** of the revised manuscript.
> Furthermore, we have added this limitation in **5. Conclusion and Future Work** of our revised manuscript, and we believe that extending our method to general datasets would be interesting future work.
>
> > **[W1.4, Q1.1, Q1.2] Impact of Selective Learning on Segmentation Map Quality and Quantitative/Qualitative Comparison**
>
> As the reviewer noted, the selective learning process does impact the quality of the generated segmentation maps.
> Following the reviewer's suggestion, we have included a new qualitative comparison in **Figure 6** of the revised manuscript.
> This comparison demonstrated that our Style-Selective LoRA significantly outperforms both DatasetDM and the original LoRA (which applies LoRA parameters to all layers).
>
> As illustrated in the figure, our Style-Selective LoRA significantly enhances the quality of segmentation maps compared to DatasetDM.
>
> Additionally, while we could not discuss this in the main paper due to the page limit, we initially included such discussion in our Supplementary **Appendix A.6 Comparison of Image-Label Alignment**.
> **Table 8** presents a quantitative comparison of image-label alignment between our Selective LoRA, DatasetDM, and other Selective LoRA variants.
> To provide quantitative results, we use the predictions from the pretrained Mask2Former model, which was fully supervised on the 100\% Cityscapes dataset and achieves a 79.40 mIoU, as a proxy for the ground truth mask.
> Below are the key results from Table 8, which compares the image-label alignment (mIoU) of DatasetDM and ours under in-domain experiments. We selected 2\% of layers for Style-Selective LoRA.
>
> |  Method   | Image-Label Alignment |
> | :-------: | :-------------------: |
> | DatasetDM |         25.18         |
> |   Ours    |         39.37         |
>
> Similar to the qualitative results, the Style-Selective LoRA outperforms both DatasetDM and the Viewpoint-Selective LoRA. We attribute this to the domain gap between the pretrained text-to-image (T2I) model (SDXL) and the source dataset (Cityscapes), as discussed in **Appendix A.6: Comparison of Image-Label Alignment** under **Analysis of the Qualitative Comparison**.
>
> > **[W1.5] Correction of Stage 4 of Figure 2 Annotation Error**
>
> We want to clarify that stage 4) of Figure 2 (in the original manuscript) is not reversed.
> Let us use the Cityscapes example as the running example.
> With Cityscapes, the style-sensitive layers learn the style of Cityscapes, while the viewpoint-sensitive layers learn the viewpoint of Cityscapes.
> This means that the style-sensitive layers can only output the style of Cityscapes, making them less effective at generating other styles of urban scenes.
> Conversely, the viewpoint-sensitive layers, which learn the viewpoint of Cityscapes, can generate diverse styles because the SD model retains knowledge of other styles.
> Therefore, to generate diverse styles, we need viewpoint-sensitive layers, and to generate diverse viewpoints, we need style-sensitive layers.
>
> We acknowledge that the terms "style sensitive" and "viewpoint sensitive" layers might be confusing due to their seemingly opposite outputs.
> To address this, we have revised **Figure 2** by removing the Style-Selective LoRA and focus on the segmentation dataset generation framework.
> Furthermore, we added qualitative results for the Viewpoint-Selective LoRA with a brief explanation in the motivation figure (**Figure 1**) to better illustrate the sensitivity.

---

> > ### Comment · Reviewer_B65c · 2024-11-24
> >
> > Thanks for authors details explanations. My concerns are resolved, thus I raise my rate.

---

> > > ### Author Response · Authors · 2024-11-24
> > >
> > > Thank you for your positive support and reassessment of our manuscript. Your constructive feedback has been invaluable in improving our work.

---

### Author Response · Authors · 2024-11-22

*We sincerely thank the reviewers for their thoughtful and constructive feedback, which has greatly contributed to improving the quality and clarity of our manuscript.*

We deeply appreciate the recognition of the strengths of our work and have refined our manuscript further by thoughtfully incorporating the suggested improvements. The strengths highlighted by the reviewers include:

- The core methodology, Selective LoRA, was acknowledged as both reasonable and novel by all reviewers (**B65c**, **hMNt**, **KMEN**, **kFPj**).
- The extensive experiments and ablation studies effectively demonstrated that the generated dataset enhances segmentation performance across various settings, including both in-domain and domain-generalization tasks (**B65c**, **hMNt**, **KMEN**, **kFPj**).
- Our work addresses the critical challenge of data scarcity, a key issue in semantic segmentation and other vision tasks (**hMNt**, **kFPj**).
- The paper is well-organized and well-written (**B65c**, **KMEN**), with a clear and concise description of the motivation (**B65c**). Additionally, the pipeline figures were noted as easy to understand (**B65c**).

We have carefully addressed all concerns raised by the reviewers and incorporated their valuable suggestions to further enhance the manuscript. Detailed responses to each reviewer’s comments are provided below.

> **Improving the overall presentation of the manuscript**

- We have added detailed explanations of the segmentation dataset framework in **Section 3.1 Overall Framework** and **Section 3.4. Training Label Generator and Generating Diverse Segmentation Datasets**.
- Further details are provided in **Appendix A.1. Implementation Details**, along with a newly added **Figure 7 Detailed Label Generator Architecture**.
- To enhance clarity, we have reorganized **Section 4. Experiments** and updated **Table 1. (previously Table 3)**.

> **Additional Experiments**: We conducted all additional experiments requested by the reviewers, as follows:

- Additional image-label alignment experiments are provided in **Figure 6** and **Appendix A.6 Comparison of Image-Label Alignment**, addressing concerns from reviewers **B65c** and **kFPj**.
- We tested our approach on a general domain dataset (Pascal-VOC), and the results are included in **Appendix A.8. In-domain Experiments for the General Domain Dataset**, as requested by reviewers **B65c** and **kFPj**.
- We conducted experiments with hand-crafted selection baselines (Selective LoRA fine-tuning on all cross-attention layers) as suggested by **hMNt**.
- Additionally, we included **InstructPix2Pix** as an additional baseline in the domain generalization setting (**Table 2**), addressing the suggestion by **KMEN**.
- To address concerns from **kFPj**, we added **Appendix A.7. Concept Sensitivity According to Prompt Augmentation**, demonstrating the robustness of our selection approach.

> **Secondary Additional Experiments**: We conducted all the experiments requested by the reviewers, which were added as part of the second round review.

- We conducted an additional image generation baseline (DATUM) in **Table 2**, as requested by **KMEN**.
- At the request of **KMEN**, we performed analyses on Image Domain Alignment and Image-Label Alignment in a domain generalization setting, detailed in **Appendix A.12. Additional Analysis of Our Generated Dataset on the Domain Generalization Setting**.
- In response to **kFPj**'s suggestion, we further analyzed class-specific performance and proposed an additional method to improve specific classes, as outlined in **Appendix A.10. Class-wise Segmentation Performance Analysis**.
- As requested by **kFPj**, we conducted new experiments on text guidance considerations, presented in **Appendix A.11. Generating Datasets with Diverse Class Names**.

We appreciate the reviewers' detailed comments and hope that our revisions satisfactorily address their concerns. Detailed responses to each reviewer's comments are provided below.

---

### Meta-Review · Area_Chair_UdKn · 2024-12-22

**Metareview:**

Dear Authors,

This draft received 6, 5, 6, 5. Some reviewers updated their scores based on author feedback however, the average rating is still lower than "marginally above the acceptance threshold". After going over the comments, draft, and feedback from authors, we find direction to be interesting but require further work.

We will encourage authors to update the draft on the basis of comments from the reviewers.


regards

AC

**Additional Comments On Reviewer Discussion:**

Authors provided additional experimental results and comments. Authors provided effective enough rebuttal that reviewers increased the rating. However, reviewers did not assign the rating above than 6, and two of them assigned 5 and 3. Later one indicated that rating could be increased to 5, therefore it was taken inconsideration.
kFPj provided further guidance too.

NOTE: hMNt  has  updated the portal to indicate rating 5. So now rating of paper is 6, 5, 6, 5. That's what was used during meta review.

---

### Decision · Program_Chairs · 2025-01-22

Reject